# GENERALIZING REASONING PROBLEMS TO LONGER LENGTHS

**Changnan Xiao**
ChangnXX.github.io
`changnanxiao@gmail.com`

**Bing Liu**
Department of Computer Science
University of Illinois Chicago
`liub@uic.edu`

## ABSTRACT

*Length generalization* (LG) is a challenging problem in learning to reason. It refers to the phenomenon that when trained on reasoning problems of smaller lengths/sizes, the model struggles with problems of larger sizes or lengths. Although it has been proven that reasoning can be learned if the intermediate reasoning steps (also known as *chain-of-thought* (*CoT*)) are given in the training data, existing studies only apply to within a given length (*interpolation*), while LG is about *extrapolation* beyond the given length. This paper begins by presenting a theorem that identifies the root cause of the LG problem, which shows why existing approaches are insufficient for LG. It then defines a class of reasoning problems for which achieving LG with Transformers can be theoretically guaranteed, provided the CoT schemes are constructed to meet a proposed condition called $(n, r)$-*consistency*. To our knowledge, limited theoretical work has been done about LG in the existing literature. In the empirical study, we introduce the CoT schemes for reasoning problems like *arithmetic*, *parity*, *addition*, *multiplication*, and *division* to train a Transformer to achieve LG for these problems.

## 1 INTRODUCTION

Large language models (LLMs) have been shown to perform reasoning tasks remarkably well (Brown et al., 2020; Liu et al., 2023; Xu et al., 2023b). However, evaluations also revealed some limitations. For example, LLMs often have difficulties in simple *addition* and *multiplication* of large numbers (Nogueira et al., 2021; Qian et al., 2022; He et al., 2024). A popular solution to improve reasoning is to use **Scratchpad** (Nye et al., 2021) or **Chain of Thought** (**CoT**) (Wei et al., 2022). Their idea is to add *intermediate steps* for each reasoning problem in the training data. For example, the training sample for calculating $3 + 2 \times 1$ may be presented as $3 + 2 \times 1 = 3 + 2 = 5$ rather than $3 + 2 \times 1 = 5$. CoT has been used to improve reasoning (Anil et al., 2022; Liu & Low, 2023). However, Dziri et al. (2023) and others reported that even with detailed CoT steps, the learned models still fail to generalize. For example, they showed that when trained with smaller problem instances, e.g., multiplication of two smaller numbers like $1234 \times 135$ based on the CoT training data, the model cannot generalize to solve larger problem instances (e.g., $235469 \times 44562$). This problem is called **length generalization** (**LG**) (Anil et al., 2022; Kazemnejad et al., 2023). Note that we distinguish **a problem**, e.g., multiplication of two numbers) and **an instance** of the problem, e.g., $1234 \times 135$.

This paper proposes *a theoretical study* of LG in learning to reason given a step-by-step CoT process for each training problem instance. We will not study the case where the CoT steps are not given but only the *direct input and output* (e.g., $3 + 2 \times 1 = 5$) are provided as it has been proven that this case isn't learnable in general (Wies et al., 2023; Feng et al., 2023; Malach, 2023). Although the learnability based on CoT has been proven for neural networks in (Wies et al., 2023; Feng et al., 2023; Malach, 2023), these studies are all under i.i.d and given problem length/size $N$. They do not cover LG. Their statements are like "*for a given dataset with training instances of a problem with length no longer than $N$, a suitable neural network can learn to solve any instances of the problem with length $N' \le N$ under a PAC upper bound.*" The key limitation of their studies is that the training problem instance length and testing problem instance length are in the same range. We will see in Sec. 4 that if the test instance length is longer than the training instance length range, the model fails badly.

This paper aims to overcome this limitation to achieve LG. **Our statement** is: "*for a given dataset with training instances of a problem with lengths no longer than $N$, learning can solve problem instances of any length $N'$ if the CoT scheme of the problem can be constructed to satisfy a proposed condition.*" This condition is called $(\boldsymbol{n}, \boldsymbol{r})$**-consistency**. Note that for the same problem, there can be many ways to provide the intermediate steps, which we call *different CoT schemes*. Some CoT schemes may fail to meet the $(n, r)$-consistency condition and struggle to achieve LG, while others may satisfy the condition and can achieve LG.

This paper makes the following contributions:

1. It first presents a theorem to identify the root cause of the LG problem, which explains why existing approaches based on various position embedding methods (Duan & Shi, 2023; Zhou et al., 2024; He et al., 2024) are insufficient for solving the LG problem.

2. It then proposes the condition $(n, r)$-*consistency*, which formally defines a *problem class* whose problems can have CoT schemes that satisfy the $(n, r)$-*consistency* condition. We prove that LG can be achieved with a Transformer for this class of problems. To our knowledge, limited theoretical analysis has been done about LG or existing approaches in the literature. In essence, the proposed method only requires the learner to see all $n$ $r$-length intervals to guarantee LG, regardless of the length of the training or testing sequences.

3. We validate the theory by using a Transformer to learn complex tasks like *arithmetic*, *parity*, *addition*, *multiplication*, and *division*, achieving LG for the tested lengths. To our knowledge, no current method attains perfect LG for multiplication for those lengths, and no reported results exist for division, which presents an even greater challenge.

## 2 RELATED WORK

Our related work includes out-of-distribution (OOD) generalization, theories about using CoT for reasoning, and empirical work on LG. We review the evaluation of reasoning of LLMs in *Appendix* A.

OOD generalization of reasoning was studied in (Abbe et al., 2023). It assumes that some value combinations are missing during training and that can result in wrong predictions on OOD data. Our work is different as we identify a condition for achieving LG. LG is a type of OOD generalization. However, there is a key difference. Since the maximal length of the training problems is always finite, a larger size/length problem can always appear in testing, which can be seen as OOD. But such an OOD is unavoidable regardless of how much data is used in training as long as it is finite. OOD generation in (Abbe et al., 2023) is solvable with more diverse training data.

Wies et al. (2023) proved that when sufficient intermediate steps (or CoT) are available, a neural network can learn any function in the **P** time complexity class. In addition, there exist functions in the **P** time complexity class that cannot be learned by any polynomial time learning algorithm without CoT. Feng et al. (2023) showed why CoT works on problems that can be decomposed into sub-problems. They also proved that it is not learnable directly without CoT. Li et al. (2023b) showed that CoT can enable the model to identify each step and then work on the step before moving to the next step in the CoT chain. Prystawski & Goodman (2023) studied why and how CoT works in LLMs. Malach (2023) proved that with some assumptions, even simple models like linear next-token predictors trained on CoT data are universal learners. However, as discussed in Sec. 1, the theorems in these papers are based on the given length/size $N$. They do not cover LG. An analysis was also done about mathematical reasoning in (Hu et al., 2024) and it shows that reasoning is done more like case-based reasoning rather than based on learned rules.

Many empirical attempts tried to modify the Transformer and learning biases to solve the LG problem better. Duan & Shi (2023), Zhou et al. (2024), Jelassi et al. (2023) and He et al. (2024) proposed different position embedding or bias calibration methods to enable the model to learn better. However, their methods are unable to solve *addition* perfectly or *multiplication* at all. Manipulating positional embeddings cannot guarantee LG because it still falls within the scope of the proposed Theorem 3.1, where the learner does not encounter positional embeddings of longer sequences in training and, therefore, cannot ensure LG in testing. Jelassi et al. (2023) further proposed to add a small number of long sequences in the training to help solve long sequences, but still could not solve *multiplication*. Chi et al. (2023) proposed a Transformer variant with weight-sharing, a working memory, etc, to improve LG for regular languages, but it still cannot solve *multiplication* or *addition*. Different

attention mechanisms and new architectures are also proposed in (Nangia & Bowman, 2018; Bowman et al., 2015; Tay et al., 2021; Chowdhury & Caragea, 2023a;b). However, they don't use CoT but only the direct input and output in training. Their methods work on various text copying and list operations but don't solve these problems and do not work on more complex large-number *addition* and *multiplication*. Theoretically, learning to reason without intermediate steps (or CoT) has been shown not learnable (Wies et al., 2023; Feng et al., 2023).

LG is also studied for text generation, which has the problem when training on short text while evaluating on longer text (Sun et al., 2023; Press et al., 2022; Ruoss et al., 2023; Han et al., 2023; He et al., 2024). However, this body of work is different than the LG problem in reasoning.

## 3 PROPOSED THEORETICAL STUDY OF LG

Let $\mathscr{S}$ be a reasoning problem and the sequence (or string) $S = s_1 \ldots s_{|S|}$ be an instance of the problem, i.e., $S \in \mathscr{S}$. We use $|S|$ to denote the length of $S$ and $S[i] = s_i$ to denote the $i$'th element of $S$, where $s_i \in V$ and $V$ is a finite vocabulary space.

A **CoT** scheme for a reasoning problem is a sequence of intermediate steps for solving the problem.[1] We use $S^t$ to denote **the input** of the $t$'th CoT step of $S$ and $S^{t+1}$ as **the output** of the step, which is also the input of the $(t+1)$'*th* CoT step. For simplicity, we use $(S^0, S^1, \ldots, S^T)$ to denote the CoT process of $S$ ($S^0$ is the same as $S$ here), where $S^T$ represents the end. Note that $T$ is a function of $S^0$, i.e. $T = T(S^0)$, but again for simplicity, we simply write $T$. For instance, given $S$ as $3 + 2 \times 1$ (an instance of the *arithmetic* problem), $S^0$ is also $3 + 2 \times 1$, $S^1$ is $3 + 2$, and $S^2$ is $5$, which form the CoT process of $S$. We use $S_k$ to represent the $k$'th instance of the problem $\mathscr{S}$, and $S_k^t$ the input of the $t$'th CoT step of $S_k$. Similarly, we use $S_k^t[i]$ to denote the $i$'th element of $S_k^t$. An element $S_k^t[i]$ can also have multiple dimensions. Below, we will also use $S_k^0$ to represent the $k$th problem instance $S_k$.

**Problem Statement of Length Generation (LG):** *For a problem $\mathscr{S}$, $\forall N > 0$, when we learn a function $\hat{f}$ that performs perfectly on the problem instances of length $\leq N$, i.e., $\hat{f}(S^t) \to S^{t+1}$, $\forall |S| \leq N$, $\forall 0 \leq t < T$,[2] we want the learned function $\hat{f}$ to also perform perfectly on problem instances of arbitrary length $N'$ (i.e., $\hat{f}(S^t) \to S^{t+1}$, $\forall N' > N$ or $N' \leq N$, $\forall |S| = N'$, $\forall 0 \leq t < T$). Note that $N'$ is of arbitrary size, and $T$ is also of arbitrary size as it is a function of $S^0$.*

We propose a condition called $(n, r)$-*consistency* that is *sufficient* for achieving LG. This defines a class of problems whose CoT schemes can be designed to satisfy the condition.[3] We prove the existence of a parameterized Transformer with a mask following the condition to achieve LG.

### 3.1 ROOT CAUSE OF THE LG PROBLEM

Before introducing the proposed condition, we present the following theorem to give the root cause of the LG problem, which also highlights what is *necessary* to resolve the problem. Without loss of generality, we represent a CoT step as a multivariate function that maps tokens to tokens. For simplicity in analysis, we assume it to be a single-valued function, mapping a set of tokens to a single token. The LG problem involves observing the function's behavior with up to $N$ tokens and predicting how it behaves with more than $N$ tokens.

**Theorem 3.1** *Define $V$ as a metric space. Denote $0 \in V$ to be the empty token. For $g_N : V^N \to [-1, 1]$, $\forall N' > N$, there exists infinitely many continuations $f_{N'} : V^{N'} \to [-1, 1]$ s.t. $f_{N'}(v_1, \ldots, v_N, 0, \ldots, 0) = g_N(v_1, \ldots, v_N)$.*

See the proof in Appendix B. Here $V$ represents the vocabulary. The number of the input tokens (the length) is represented by $N$. Let the *ground truth* function reasoning on *arbitrarily many* tokens be $g_\infty : \prod_{i=1}^{\infty} V \to [-1, 1]$. The *ground truth* function reasoning on up to $N$ tokens is represented by $g_N$, which restricts $g_\infty$ onto the first $N$ tokens, i.e. $g_N(v_1, \ldots, v_N) = g_\infty(v_1, \ldots, v_N, 0, 0, 0, \ldots)$.

---

[1] There can be many CoT schemes for the same problem as we discussed earlier and will see shortly.

[2] Notice the slight difference from the existing CoT modeling, which is $\hat{f}(S^0) = S^1, S^2, ..., S^T$.

[3] This does not claim that $(n, r)$-consistency is necessary for achieving LG or that it is possible to design CoT schemes for all reasoning problems to satisfy $(n, r)$-consistency.

Theorem 3.1 states that there exists an infinite number of continuation $f_{N'}$ of a higher dimension $N'$ that can achieve the effect of the function $g_N$ of a lower dimension $N$.

The LG problem is to attain $g_{N'}$ ($N' > N$) when observing $g_N$, which is the constraint that $g_{N'}$ restricted on $N$ dimensions is $g_N$. However, Theorem 3.1 shows that there exist infinitely many functions $f_{N'}$ that can produce the effect of $g_N$ when restricted on $N$ dimensions. This means that, with only the continuity bias and the input and output data of $g_N$, it is **almost impossible to predict** the correct $g_{N'}$ as there exist infinitely many $f_{N'}$ (which are not $g_{N'}$) that satisfy the same constraint.

This analysis shows that with only the input and output data of $g_N$, it is insufficient to achieve LG, i.e. predicting $g_{N'}$. We have the following ***necessary*** condition for achieving LG for a problem:

- *Sufficient bias needs to be introduced to the problem s.t. with the bias, $f_{N'}$ in Theorem 3.1 is made equal to $g_{N'}$ uniquely (or with high concentration).*[4]

In what follows, we propose a specific bias, namely the $(n, r)$-*consistency*, which is sufficient to achieve LG. This bias is pattern-based and requires the consistency condition to ensure that $g_{N'}$ is uniquely determined when its pattern matches the corresponding pattern in $g_N$.

## 3.2 $(\boldsymbol{n}, \boldsymbol{r})$-CONSISTENCY: THE INTUITIVE IDEA

We now intuitively discuss the proposed $(n, r)$-consistency. Denote $\mathscr{S}$ as a problem and $S^0 \in \mathscr{S}$ as an instance of the problem. In a CoT step $t$, we have the input $S^t$ and output $S^{t+1}$. Our goal is to find a solver (or learn a function) $f$ that predicts $S^{t+1}$ given $S^t$, i.e. $S^{t+1} = f(S^t)$.[5] The problem of predicting $S^{t+1}$ can be decomposed into sub-problems of predicting each element of $S^{t+1}$, i.e., $S^{t+1}[i]$ for each position index $i$. To do so, we further ensure that the lengths $S^t$ and $S^{t+1}$ are the same by adding blank space elements in suitable positions. $(n, r)$ basically means a context with $n$ $r$-length intervals (or subsequences) in $S^t$ can predict $S^{t+1}$.

For example, let us consider one instance, `123+567`, of the problem, *addition*. We consider the CoT step with the input $S^0 = $ '123+567= \$0' and output $S^1 = $ '123+567=?90', where ? indicates that 0 is carried and \$ indicates that 1 is carried. To ensure that the input and the output are of the same length, a blank space (or empty) element is added after '=' in $S^0$ in the CoT scheme. We call this CoT scheme of addition as ***addition*-[1]**.

Let us consider the prediction of the 10's element 9 in the output $S^1$, i.e., $S^1[10] = 9$. The 10's position of $S^0$ is \$. Let us say that we use the three elements ***interval*** or subsequence ' \$0' in $S^0$ (i.e., $r = 3$ and \$ is the *central element* of the interval) as the context to predict 9 in the 10's position of $S^1$, which is clearly insufficient as 9 can only be calculated by using 2, 6, and \$ in $S^0$. This means that we need at least three intervals in $S^0$ as a context to predict 9 in $S^1$, i.e., '123', '567', and ' \$0', where 2, 6, and \$ are central elements in the intervals respectively, and then $n = 3$. We call ' \$0' the ***anchor interval*** as the position (10) of its central element \$ is the one that we want to predict in $S^1$ with this set of intervals, which we call a $(3, 3)$-***context*** for predicting the value (i.e., 9) in the 10's position of $S^1$. $(3, 3)$-***context*** represents three 3-length intervals.

We now introduce the concept of $(n, r)$-*consistency* with regard to a problem, $(3, 3)$-consistency with regard to addition in our case above. **Consistency** here means that 9 should be predicted for the position of the central element of the anchor interval (the position of \$ in ' \$0' in $S^0$) whenever the three intervals (' \$0': '123', '567') appear in an instance of the *addition* problem. Note that we put the anchor interval first and separate it from the other intervals using ':' to indicate what we want to predict. This $(3, 3)$-context is *not consistent* for the *addition* problem. This is because we can easily find another problem instance that contains the $(3, 3)$-context but does not predict '9' for the position of the central element of the anchor interval. For example, the input $S^0 = $ '12342+45678= \$0' and output $S^1 = $ '12342+45678=\$20'. Obviously, $S^0$ here contains the above $(3, 3)$-context, but the prediction for the position of the central element of the anchor interval (the position of '\$') in the input $S^0$ should be **2** in the output $S^1$, not 9. These two scenarios result in a **conflict** in learning. Thus, addition-[1-line] is not $(3, 3)$-*consistent* for $n = 3$ and $r = 3$. The intuition of $(n, r)$-*consistency* is that we want to learn and predict the same element value with the same context with no conflict or uncertainty.

---

[4]We still don't have a complete set of biases contributing to the *necessary* condition. This paper provides one such bias, i.e., $(n, r)$-consistency, that is sufficient for achieving LG.

[5]For simplicity, we omit $N$ in $f_N$, where $N$ represents the length of the problem instance.

We can design a different CoT scheme for addition using two lines to achieve $(3,3)$-*consistent*. We call this scheme **addition-[2]**, which uses tags to indicate the digits to be calculated next (see the CoT process in Fig 3 of *Appendix* F). The two CoT examples above become,

$$123 + 567 = \ \$0 \Rightarrow \begin{pmatrix} 123 + 567 = & \$0 \\ I & J & K \end{pmatrix}, \ and$$

$$12342 + 45678 = \ \$0 \Rightarrow \begin{pmatrix} 12342 + 45678 = & \$0 \\ I & J & K \end{pmatrix},$$

where $I$ and $J$ indicate the digits to be added next and $K$ indicates the position of the output. In this case, each element has 2 dimensions and when the second dimension is not $I$, $J$, or $K$, it is an empty token, which is also significant. In this 2-dimensional case, the corresponding $(3,3)$-context of $S^0 = $ '123+567= $0' becomes (notice the anchor interval in the first position),

$$\begin{pmatrix} `\$0' : `123', `567' \\ K & I & J \end{pmatrix}.$$

In this case, the $(3,3)$-context is not contained in the above CoT step of $12342 + 45678$. In fact, no other possible problem instance can have the same three 2-dimensional intervals whose central elements are not the elements to be calculated next because $I$, $J$, and $K$ indicate the elements to be calculated next. Thus, *addition*-[2] is $(3,3)$-*consistent*, meaning that for any instance of the *addition* problem if it contains the above $(3,3)$-context, the output element at position $K$ will always be 9.

To summarize, the $(3,3)$-context example implies that (1) a context is independent of the positional distances between any pair of intervals in it and (2) the same or consistent output is obtained if any CoT step input of a problem instance contains the same context. Thus, this example introduces a bias that if a local context in the input always implies the same output without requiring any dynamic information depending on the length or position (e.g., position encoding in the Transformer), it's possible to achieve LG. Based on this idea, we propose $(n,r)$-consistency and show it is one special bias that is *sufficient* to achieve LG by a parameterized Transformer.

### 3.3 $(n,r)$-CONSISTENCY: DEFINITION AND PROPERTIES

We now formally define $(n,r)$-*consistency*. For simplicity of notations, we will use $S^0$ to represent $S^t$ and $S^1$ to represent $S^{t+1}$. Similarly, we can also call any $S^t$ a problem instance. We now introduce the concept of **consistent context**.

**Definition 3.2 (Consistent Context)** *Denote a context $h = (a_1 : a_2, \ldots, a_{|h|})$ as a sequence of $|h|$ intervals, where $a_1$ is called the **anchor interval** and each interval consists of a sequence of $r$ elements, i.e. $a_j = s_{j,1} \ldots s_{j,r}, s_{j,l} \in V, 1 \le j \le |h|, 1 \le l \le r$, where $V$ is a finite vocabulary.[6]*

*(i) For a problem instance $S^0 = s_1 \ldots s_{|S^0|}$, we say $h \sqsubset S^0$ (contained in) if there exists $1 \le m_1, \ldots, m_{|h|} \le |S^0|,$[7] s.t. $s_{m_j} \ldots s_{m_j+r-1} = a_j, 1 \le j \le |h|$.*

*(ii) We say $h$ is a context of a problem $\mathscr{S}$ if there exists a problem instance $S^0 \in \mathscr{S}$ s.t. $h \sqsubset S^0$.*

*(iii) We define $c(h, S^0) = m_1 + \lfloor \frac{r}{2} \rfloor$ as the position index of the central element of the anchor interval $a_1 = s_{m_1} \ldots s_{m_1+r-1}$ in $S^0$ (due to the flooring, $r$ does not have to be **odd**). For example, given $r = 3$, $S^0 = 1 + 2 \times 1$ and $a_1 = 2 \times 1$, then $m_1 = 3, \lfloor \frac{r}{2} \rfloor = 1, c(h, S^0) = 4$, and $S^0[c(h, S^0)] = \times$.*

*(iv) We say $h$ is a **consistent context** if for any $S_1^0, S_2^0 \in \mathscr{S}$ and their respective CoT step outputs $S_1^1, S_2^1$, when $h \sqsubset S_1^0, h \sqsubset S_2^0$, they always have the same element at the central element position of $a_1$ of $h$, i.e., $S_1^1[c(h, S_1^0)] = S_2^1[c(h, S_2^0)]$. To ensure the equal length of $S_k^0$ and $S_k^1$, blank space elements are inserted in suitable positions (see below).*

*(v) For a consistent context $h$, define $\psi(h) = S^1[c(h, S^0)], \forall S^0 \in \mathscr{S}$ s.t. $h \sqsubset S^0$.*

When $h = (a_1 : a_2, \ldots, a_{|h|})$, $h' = (a_1' : a_2', \ldots, a_{|h'|}'))$, $|h'| > |h|$ and $a_i = a_i', 1 \le i \le |h|$, we say $h$ is the *prefix* of $h'$ and write $h \prec h'$. We write $h \preceq h'$, when $h \prec h'$ or $h = h'$.

---

[6] It is not important where the intervals other than the anchor interval are located.

[7] It is not required that $m_1 < m_2 < \cdots < m_{|h|}$ and every $a_j$ in $h = (a_1 : a_2, \ldots, a_{|h|})$ doesn't contain any positional information in $S^0$.

Definition 3.2 constrains only the anchor interval $a_1$ of $h$ and it seems to leave the other intervals unconstrained except $h$ is contained in $S^0$. As indicated in Sec. 3.2, the other intervals usually contain the other elements involved in a reasoning step. This leads to our $(n, r)$-consistency definition.

**Definition 3.3 (($n, r$)-Consistent Problem)** *We say a problem $\mathscr{S}$ is $(n, r)$-consistent, if for every problem instance $S^0$ in $\mathscr{S}$, i.e., $\forall\, S^0 \in \mathscr{S}$, denote $S^1$ as the CoT step output of $S^0$, for every index or position $i$ of $S^1$ (i.e. $\forall\, 1 \leq i \leq |S^1|$), there exists a consistent context $h_{[i]}$ s.t. (i) $c(h_{[i]}, S^0) = i$, (ii) $|h_{[i]}| \leq n$, (iii) all intervals in $h_{[i]}$ is $r$-length.*

For simplicity, we omit (ii) and (iii) of Def. 3.3 in the following discussions unless necessary. When $\mathscr{S}$ is $(n, r)$-consistent, for $S^0 \in \mathscr{S}$, determining the output $S^1$ of the CoT step can be decomposed into determining every element $S^1[i]$ for every index $i$. Given a particular index $i$, if we can find some consistent context $h_{[i]}$ s.t. $c(h_{[i]}, S^0) = i$, where the existence of $h_{[i]}$ is guaranteed by Def. 3.3, then we have $S^1[i] = \psi(h_{[i]})$ by Def. 3.2. By concatenating $S^1[i]$'s, we achieve the objective of determining $S^1$. Intuitively, as discussed in Sec. 3.2, we want the CoT scheme to have a consistent context $h_{[i]}$ to predict the element $S^1[i]$ in each output position $i$ with no uncertainty.

Both the consistency property of $h$ and the $(n, r)$-consistency property of $\mathscr{S}$ are monotonic.

**Property 3.4** *If $h$ is a consistent context and $h \preceq h'$, then $h'$ is also a consistent context.*

**Property 3.5** *If $\mathscr{S}$ is $(n, r)$-consistent, then for $\forall\, n' \geq n$, $\forall\, r' \geq r$, $\mathscr{S}$ is $(n', r')$-consistent.*

The proofs are given in Appendix B. It is easy to see that different intervals in the same context $h$ may have different lengths (number of tokens). We use the same size $r$ in our theory for simplicity. It is also important to note that in practice, when we design a CoT scheme, we don't need to design a minimal consistent context. It's enough as long as the context is consistent.

### 3.4  $(n, r)$-Consistency Implies Length Generalization (LG)

We now show that when a problem is $(n, r)$-consistent, LG can be achieved (see the problem statement at the beginning of Sec. 3). That is, there always exists a solver (e.g., model or algorithm) that can learn to predict the CoT step output for an arbitrary problem instance of any length.

Here we propose one solver that can achieve LG. It is a Transformer-based model. The model takes a problem instance as the input and output of the CoT step of the problem instance. We show that there exists a proper parameterization of the model, s.t. the CoT step output prediction is correct for input problem instances of arbitrary length.

**Theorem 3.6** *If a problem $\mathscr{S}$ is $(n, r)$-consistent, then there exists an $n$-layer Transformer model $f$, for $\forall\, S^0 \in \mathscr{S}$ with CoT process $(S^0, \ldots, S^T)$, we always have $S^{t+1} = f(S^t), 0 \leq t < T$.*

**[Proof Sketch.]** For $\forall\, S^0 \in \mathscr{S}$ with $(S^0, \ldots, S^T)$, for $\forall\, 0 \leq t < T$, we construct a $n$-layer Transformer model $f$ s.t. $S^{t+1} = f(S^t)$. For simplicity, we denote $S^t$ as $S^0$ and $S^{t+1}$ as $S^1$. Since $\mathscr{S}$ is $(n, r)$-consistent, for each $1 \leq i \leq |S^1|$, there exists a consistent context $h_{[i]}$ s.t. $c(h_{[i]}, S^0) = i$. The key idea to construct $f$ is to extract $h_{[i]}$ for each $i$, so that $S^1[i] = \psi(h_{[i]})$ is determined. The 1st layer applies a local padding mask with relative position encoding, where the local mask guarantees that the token at index $i$ can only observe the interval that centers at $i$. The 1st attention layer aggregates $a_1$ for each $h_{[i]}$'s. The 1st feed-forward layer lists the contexts that are potentially consistent when concatenating $a_1$ to the end. The 2nd attention layer has no mask, which seeks for potential $a_2$'s that $(a_1, a_2)$ is the prefix of some consistent context $h$. The 2nd feed-forward layer produces an indicator if $(a_1, a_2)$ is already a consistent context. The process continues. Since $\mathscr{S}$ is $(n, r)$-consistent, there must exists a consistent $h_{[i]}$ for each $i$ after $n$ layers. The last layer maps $h_{[i]}$ to $\psi(h_{[i]})$ for each $i$.

The **full proof** is given in Appendix B. This theorem shows $(n, r)$-consistency implies the existence of a Transformer model that can achieve LG.[8] The interplay of the problem complexity and the network capacity (e.g., the number of parameters and layers) is discussed as part of the proof.

---

[8]This does not imply that $(n, r)$-consistency guarantees the LG learnability of the Transformer. However, while our theory only establishes that $(n, r)$-consistency implies the existence of a Transformer model capable

## 3.5 How to Design CoT Schemes to Achieve $(n, r)$-consistency

It's challenging to create a universal procedure that can generate CoT schemes across reasoning problems to satisfy $(n, r)$-consistency. Each problem is likely to require a tailored method as different problems have entirely different computation steps. However, as demonstrated by the examples in Sec. 3.2 and those in the experiment section (Sec. 4), the general approach is quite clear. We use tags to inform the learner what positions or elements are involved in calculating each CoT step, which naturally creates consistent local contexts to ensure that each output element in a reasoning step can be learned and predicted without conflicts across all possible instances of the problem. For example, the tags $I$, $J$, and $K$ in the addition problem in Sec. 3.2 tell the learner about which digit ($I$) from one number should be added to which digit ($J$) from the second number, and what the result should be in the output position ($K$) in each reasoning step. In a nutshell, the tags implicitly inform the learner how the calculations are done in solving the problem. With such tags, the learner can learn the underlying reasoning rules without ambiguity or conflict. The $(n, r)$-consistency condition checks whether the provided tag information is sufficient for learning to achieve LG. The key weakness of existing approaches is that they learn short-cut patterns or regularities, rather than reasoning rules. That is why they cannot extrapolate to longer lengths. To achieve LG, the learner must learn second-order patterns (reasoning or calculation rules for each problem) similar to how we teach children to do addition or multiplication by making them learn the rules of calculation rather than guess based on example inputs and outputs alone.

However, defining tags manually and then checking $(n, r)$-consistency for each possible problem is not scalable. There exists an automatic method that can transform a problem into $(n, r)$-consistent, which is induced by Def. 3.3 and outlined in *Appendix* C. However, its high computational complexity makes it impractical. Improving this automated CoT design method is an interesting future work.[9]

## 4 Experiments

Our experiments verify (1) for a CoT scheme of a problem, if it is $(n, r)$-consistent, it is solvable for LG, and (2) for the same problem, one CoT scheme may not be solvable for LG, but another may. The ***code*** of our system can be downloaded at https://openreview.net/forum?id=zpENPcQSj1.

### 4.1 Experimental Problems, CoT schemes, and $(n, r)$-Consistency

**Experimental Problems.** We use 5 reasoning problems in our experiment. **(1) *arithmetic* in $F_7$** (the finite prime field with seven elements, i.e., $F_7 = \{0, 1, 2, 3, 4, 5, 6\}$), where the calculations are under the sense of '*mod 7*', **(2) *parity***, which is the problem of deciding whether there are an *even* or *odd* number of 1's in a sequence of 0's and 1's, **(3) *addition*** of two integer numbers, and **(4) *multiplication*** of two integer numbers. **(5) *division*** of two integer numbers.

**CoT Schemes for the Problems.** As mentioned earlier, each CoT step is a pair (Input[i], Output[i]). Fig. 3 in *Appendix* F gives an example for each CoT scheme. Below, we detail the CoT schemes.

**(1) *arithmetic* in $F_7$.** It is formulated in the usual way and represented as input and output pairs. A detailed training example is given in Fig. 3 in *Appendix* F. This CoT scheme achieves LG, as it is $(1, 17)$-consistent. Each element belongs to at most one calculation step, and the distance between the elements that are calculated together is at most 4, e.g., between '(' and ')' in $(3 + 2)$. Therefore, to determine whether the $i$'th element might be calculated next, we consider $i$'th neighbors of radius 4, i.e. $s_{i-4} \ldots s_{i+4}$. To check whether $s_{i-4}, \ldots, s_{i+4}$ are in other higher priority calculations, we further consider each neighbor of radius 4, i.e. $s_{i-8} \ldots s_{i+8}$. Therefore, every 17-length interval determines the next central element, regardless of problem instance and position index. Note that we

---

of achieving LG, our experiments show that Transformer-based models can learn to achieve LG for challenging mathematical reasoning tasks. This suggests that $(n, r)$-consistency may lead to a stronger conclusion, such as the learnability of a Transformer-based model capable of achieving LG or extrapolation, beyond the traditional i.i.d. assumption. We leave this for future investigation.

[9]That being said, we envision an approach in which we only need to design CoT schemes for a finite set of basic reasoning functions. More complex reasoning tasks are compositions of these basic functions. When addressing a complex problem, the system call upon the relevant basic functions as needed. This approach may reflect how humans learn and reason. Relying on ever-increasing amounts of sequential data to learn short-cut patterns in a brute-force manner may not be the most effective or efficient solution.

Table 1: Experimental settings. Train Length: Training length. LG Test $i$: Length generalization test with longer lengths. We have one training set and **6 test sets** for each problem. Test set 0 has the same length setting as the Training Length setting and is thus omitted from this table. $L$ stands for length. For *addition*-[1/2] $(a + b)$, *multiplication*-[1/11] $(a \times b)$ and *division*-[12] $(c \div a = b)$, $L$ is the number of digits in $a$ or $b$.

|  | Train Length | LG Test 1 | LG Test 2 | LG Test 3 | LG Test 4 | LG Test 5 |
|---|---|---|---|---|---|---|
| arithmetic in $F_7$ | $L \in [3, 20)$ | $L \in [3, 30)$ | $L \in [3, 40)$ | $L \in [3, 50)$ | $L \in [3, 60)$ | $L \in [3, 100)$ |
| parity-[2] | $L \in [1, 8)$ | $L \in [1, 30)$ | $L \in [1, 40)$ | $L \in [1, 50)$ | $L \in [1, 60)$ | $L \in [1, 100)$ |
| addition-[1/2] | $L \in [1, 8)$ | $L \in [1, 9)$ | $L \in [1, 10)$ | $L \in [1, 11)$ | $L \in [1, 16)$ | $L \in [1, 21)$ |
| multiplication-[1/11] | $L \in [1, 6)$ | $L \in [1, 7)$ | $L \in [1, 8)$ | $L \in [1, 9)$ | $L \in [1, 10)$ | $L \in [1, 11)$ |
| division-[12] | $L \in [1, 6)$ | $L \in [1, 7)$ | $L \in [1, 8)$ | $L \in [1, 9)$ | $L \in [1, 10)$ | $L \in [1, 11)$ |

pad blank or empty elements in the training data to ensure that every element in any problem instance $S^0$ can be a center element in a context of length 17.

**(2) *parity*.** It is formulated in 2 lines, i.e., *parity*-[2]. On the 2nd line, ? indicates the current position of the CoT process, 1 represents *odd* and 0 represents *even*. See Fig. 3 in *Appendix* F for an example. This CoT scheme can achieve LG as it is $(1, 2)$-consistent. For any problem instance $S^0$, any position/index $i$, if $c(h, S^0) = i$, where $h = (a_1)$ and $a_1 = s_1 s_2$, then (1) when ? is not in $a_1$, $S^1[i] = s_2$, (2) when ? is in $s_1$ or $s_2$, it's not difficult to check that $S^1[i]$ is determined by $a_1$ without ambiguity. Therefore, every 2-length interval context determines an output central element, regardless of the problem instance and position index.

**(3) *addition*.** It is formulated in CoT in two ways: *addition*-[1] and *addition*-[2]. As discussed in Sec. 3.2, *addition*-[1] is not $(n, r)$-consistent, and *addition*-[2] is $(3, 3)$-consistent. See Fig. 3 in *Appendix* F for a full training example for each formulation in the CoT input and output format.

**(4) *multiplication*.** It is formulated in two ways: *multiplication*-[1] (one line) and *multiplication*-[11] (11 lines). *multiplication*-[1] is not $(n, r)$-consistent, but *multiplication*-[11] is $(12, 3)$-consistent.

For *multiplication*-[1], we decompose the problem into two stages. In the 1st stage, we transform multiplication into a summation of multiple integers. In the 2nd stage, we solve the summation recursively. An example is shown in Fig. 3 in *Appendix* F. The 2nd stage is not $(n, r)$-consistent as *addition*-[1] is not. The 1st stage is also not $(n, r)$-*consistent*. For instance, let input[k] = '$a \times b = \underbrace{a + \cdots + a}_{k} + ?$'. When $k < b - 1$, output[k] = '$a \times b = \underbrace{a + \cdots + a}_{k+1} + ?$'. when $k = b - 1$, output[k] = '$a \times b = \underbrace{a + \cdots + a}_{k+1}$'. In this example, whether to add '$+?$' or to go to the second stage depends on $b$ and the number of existing $a$'s. For $\forall (n, r)$, there always exists a large enough $a$ and $b$ s.t. $n$ $r$-length intervals cannot cover all $a$'s as well as distinguishing all $a$'s. It's not difficult to exploit this fact to construct inconsistent instances for arbitrary context $h$.

For *multiplication*-[11] (each element has 11-dimensions), an example is given in Fig. 3 in *Appendix* F. When calculating $a \times b$, since *addition*-[2] solves LG, we only need to multiply each digit of $a$ and each digit of $b$ and add the product to the result by merging *addition*-[2] into the CoT process.

*Multiplication*-[11] is $(12, 3)$-consistent. For any $h = (a_1 : a_2, \ldots, a_{12})$, (1) when $a_1$ doesn't contain any indicator, the output of the central element of $a_1$ is simply identical to the input, (2) when $a_1$ contains at least one of the indicators, by letting $a_2, \ldots, a_{12}$ containing other 11 indicators at the center, it's not difficult to find that the output of the central element of $a_1$ is determined by context $h$.

**(5) *division*.** It is formulated in 12 lines, i.e. *division*-[12]. It's almost the same as *multiplication*-[11] except that multiplication does addition but division does subtraction. The numerator may fail to subtract the denominator (numerator is smaller than the denominator) and the numerator has to be restored before the subtraction. An example is given Fig. 3 in *Appendix* F. For the same reason as *multiplication*-[11], *division*-[12] is $(10, 3)$-consistent.

### 4.2 DATA GENERATION

**Training Set.** The model for each problem is trained with a training set of $50k$ batches. Each batch contains 256 CoT steps. For each problem, we first randomly generate an instance of the problem and then its detailed CoT steps. Each CoT step is a pair (Input[i], Output[i]), as shown in Fig. 3 in

*Appendix* F. We put the CoT steps of each generated problem instance into a batch until it reaches 256. The steps of the last problem instance that overflow the 256 batch go to the next batch.

**Test Sets.** We use 6 test sets to evaluate the model learned for each problem. The 5 columns marked 'LG Test $i$' in Table 1 give the length ranges of the 5 test sets for each problem, where the maximum lengths of the test sets increase gradually. The first test set has the same length range as that of the training set and thus shares the 'Train Length' column. Every test set consists of $1k$ questions (test problem instances), which are in sequence format with no CoT steps, e.g., $3 + 2 \times 2$.

**Training and Test Data Generation.** Every training or test set is generated independently. The training set and each test set are generated in the same way for each problem except that for the training set, we also need to generate its CoT steps for each problem instance based on individual CoT schemes, but for each test set, we do not. For each training instance, the CoT steps end with an "EOS" token. The additional data generation details are as follows:

**(1) arithmetic in $F_7$.** The length (number of elements $L$) of a problem instance in the dataset is generated to be as close to the maximum length as possible (see Table 1).

**(2) parity.** The length $L$ (number of elements) of a problem instance is uniformly sampled from 1 to the maximum length (see Table 1). Each element in the sequence is sampled randomly from $\{0, 1\}$.

**(3) addition.** The number of digits (or length $L$) in $a$ or $b$ as in $a + b$ of each problem instance is uniformly sampled from 1 to the maximum length (see Table 1). Each digit in $a$ or $b$ is sampled from $\{0, 1, 2, 3, 4, 5, 6, 7, 8, 9\}$, while the left-most digit is removed when it's 0.

**(4) multiplication.** The number of digits (or length $L$) in $a$ or $b$ as in $a \times b$ of each problem instance is uniformly sampled from 1 to the maximum length (see Table 1). Each digit in $a$ or $b$ is also sampled from $\{0, 1, 2, 3, 4, 5, 6, 7, 8, 9\}$, while the left-most digit is removed if it is 0.

**(5) division.** The number of digits (or length $L$) in $a$ or $b$ as in $c \div a = b$ of each problem instance is uniformly sampled from 1 to the maximum length (see Table 1). Each digit in $a$ or $b$ is also sampled from $\{0, 1, 2, 3, 4, 5, 6, 7, 8, 9\}$, while the left-most digit is removed when it is 0. Then $c = a \times b$ and $a$ are given as numerator and denominator, and $b$ is the correct answer. This setup ensures that the answer $b$ is an integer. But our formulation can be easily extended to allow decimal answers.

### 4.3 IMPLEMENTATION DETAILS

The models for *arithmetic*, *parity*, *addition*-[1], and *multiplication*-[1] have 3 Transformer encoders with relative position encoding. The models for *addition*-[2], *multiplication*-[11] and *division*-[12] have 6 Transformer encoders, as they are $(n, r)$-consistent and $n > 1$. By the proof of Thm. 3.6, both the attention layer with relative position encoding and the attention layer without position encoding are needed. The 1st, 3rd, and 5th encoders use relative position encoding with padding masks, which for the $i$'th token is $\{j | \frac{r}{2} \leq j - i < \frac{r}{2}\}$, i.e. the $r$-length interval. The 2nd, 4th, and 6th encoders don't use position encoding, which exchanges information of the $n$ intervals. The optimizer is Adam and the learning rate is 0.0001. The training data for each task contains 12.8M CoT steps. Due to the complexity of *multiplication* and *division*, they are additionally trained on 25.6M CoT steps with learning rate 0.000005.

When a CoT scheme has multiple lines, e.g., *multiplication*-[11], each element has multiple dimensions and each dimension has a token, including the blank/empty token. The model first maps each token into an embedding vector and then concatenates the vectors at the same position index. Then a fully connected layer maps the concatenated vector into a vector via a linear transformation. The remaining model is a standard Transformer with masks and position encoding described above.

We pad empty tokens ' ' at the beginning and/or at the end to guarantee that each position (or element) can be the central element of a sequence or interval of length $r$ if the problem formulation is $(n, r)$-consistent. Specifically, we pad $\lfloor \frac{r}{2} \rfloor$ empty tokens at the beginning and $\lfloor \frac{r-1}{2} \rfloor$ tokens at the end. For *addition*-[2], we further pad an empty token on the right of $=$ to align input and output (see Sec. 3.2 and Fig. 3 in *Appendix* F). For CoT schemes that are not $(n, r)$-consistent, we don't pad as it is unclear where and how many empty tokens to pad.

**Training:** As mentioned above, the model for each problem is trained with $50k$ batches and each batch contains 256 CoT steps. The training is done only in one epoch.

Figure 1: Test results in accuracy.

**Testing:** We solve each test question using the trained model. The output generation stops when the termination token "EOS" is predicted.

For all problems, the final output is considered correct only when it is identical to the ground truth. The accuracy is the number of correctly answered instances divided by the total number of instances.

## 4.4 EXPERIMENTAL RESULTS

**arithmetic in** $F_7$. Fig. 1 shows the problem achieves 100% accuracy for all test sets as the problem is $(1, 17)$-consistent.

**parity**. It is formulated in 2 lines (*parity*-[2]), and achieves 100% accuracy in LG for all test sets (see Fig. 1) as the problem is $(1, 2)$-consistent.

**addition**. It is formulated in two ways: *addition*-[1] and *addition*-[2]. Fig. 1 shows that *addition*-[1] fails to achieve LG as it is not $(n, r)$-consistent. But *addition*-[2] achieves 100% accuracy for all test sets (Fig. 1) as it's $(3, 3)$-consistent (Sec. 3.2).

**multiplication**. It is formulated in two ways: *multiplication*-[1] and *multiplication*-[11]. Since *multiplication*-[1] is not $(n, r)$-consistent, Fig. 1 shows poor LG accuracy. Since *multiplication*-[11] is $(12, 3)$-consistent, it is solvable for LG with 100% accuracy for all test sets (Fig. 1).

**division.** It is formulated as *division*-[12]. Since it's $(10, 3)$-consistent, it achieves LG, but there are three cases where we do not achieve 100% accuracy (Fig. 1). See the details in *Appendix* D.

*Appendix* D reports additional experimental results for these problems with even longer lengths to stress test the system, where we will see imperfections. We will also explain why they happen even when the $(n, r)$-consistency condition is satisfied for these problems.

Note that we don't compare with existing systems because, as we discussed in Sec. 2, no existing reported system can solve multiplication and no reported system has even attempted division.

## 5 CONCLUSION

Length generalization (LG) is a challenging problem in learning reasoning skills. There is little theoretical understanding so far. This paper introduces a theoretical framework for understanding reasoning that extends beyond specific problems or architectures. It first introduced a theorem to show the root cause of the LG problem, which explains why existing approaches are insufficient for solving the LG problem. It then formally defines a class of problems for which achieving LG with Transformers can be theoretically proven. Specifically, the problem class's CoT schemes can be designed to satisfy a condition called $(n, r)$-consistency. Empirical results align well with the theory, showing perfect LG for large tested lengths on several challenging reasoning problems.

*Limitations and future work*: The proposed $(n, r)$-consistency can be seen as a sufficient condition for achieving LG. However, regarding necessary conditions, we only know that sufficient learning biases are needed, but not the complete set of biases contributing to the necessary condition. Further research is needed. In the current approach, we need to manually design a CoT scheme to satisfy the $(n, r)$-consistency for a reasoning problem. A future research direction is to automate the design of CoT schemes and realize the approach outlined in footnote 9.

ACKNOWLEDGMENTS

Bing Liu's work was supported in part by three NSF grants (IIS-2229876, IIS-1838770, and CNS-2225427).

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

**Appendix**

# A   RELATED EMPIRICAL WORK

Here we review the related empirical work, which includes the evaluation of LLMs in reasoning, chain-of-thoughts for reasoning, dealing with length generalization (LG) in learning to reason, and dealing with LG in text generation.

**Evaluations and limitations of LLMs in reasoning**. Continuing with the discussion about evaluations of the reasoning capabilities of LLMs in Sec. 1, we present a more extensive literature survey here. In general, evaluations conducted on several latest LLMs showed that they struggled with many reasoning tasks Gendron et al. (2023); Tang et al. (2023).

In Sec. 1, we discussed empirical works about LG Anil et al. (2022); Dziri et al. (2023); Zhang et al. (2022). In these papers, the authors also tried to mitigate the problem through improved training and CoT Anil et al. (2022), improved prompting and fine-tuning of LLMs Zhang et al. (2022), and curriculum learning Abbe et al. (2023). An evaluation of the deductive reasoning capability of LLMs was also conducted in Prystawski & Goodman (2023), which shows that CoT helps improve the results, but does not achieve perfect accuracy. None of them studied the LG problem theoretically as we do. Below, we focus on surveying other empirical works. Many of them identified limitations of LLMs in solving different reasoning problems, but few have characterized the limitations in a formal manner to facilitate theoretical investigation.

Meadows et al. (2023) created a dataset specifically for mathematical reasoning that can be perturbed. They showed that perturbations of the tasks heavily affect the results, reducing F1 score from 97% to 17%, which suggests that inference is likely to be dominated by surface-level patterns unrelated to the deeper understanding of the mathematical operators. However, this evaluation was done using only BERT Devlin et al. (2018) based models, but not on more recent LLMs like ChatGPT and GPT4. Wu et al. (2023) used "counterfactual" tasks that deviate from the standard reasoning tasks to evaluate LLMs. It was found that the performance degrades substantially compared to the default conditions, which again suggests that while LLMs can perform reasoning to some extent, they often rely on narrow, non-transferable procedures or surface patterns for task-solving. A counterfactual-based evaluation was also done in (Li et al., 2023a), which reached the same conclusion.

Liu et al. (2023) evaluated ChatGPT and GPT-4 on logical reasoning. The results showed that they do relatively well on well-known public domain datasets, but their performances drop substantially when newly released and out-of-distribution datasets are used. Xu et al. (2023b) also evaluated LLMs using logical reasoning (deductive, inductive, abductive, and mixed-form reasoning) and gave pros and cons of LLMs. She et al. (2023) created a dataset for reasoning involving negations and evaluated LLMs and showed poor results. Ando et al. (2023) created a dataset, originally designed for psychological experiments to assess human logical abilities in syllogistic reasoning. The authors examined three types of biases observed in human syllogistic reasoning: *belief biases*, *conversion errors*, and *atmosphere effects*. The evaluation on LLMs showed that they struggle with problems involving these biases too. Tan et al. (2023) created a dataset to evaluate LLMs on temporal reasoning and showed some weaknesses of LLMs. They then proposed an approach to improve the results.

**Chain of thoughts (CoT) and variants**. Earlier prompting for solving reasoning problems using LLMs only states the question and the answer. They found that these two pieces of information are insufficient for LLMs to learn to perform effective reasoning. Then *chain of thought* (CoT) prompting (Wei et al., 2022) was proposed to improve the situation. CoT basically contains the detailed intermediate reasoning steps between the question and the answer for fine-tuning the LLMs, which significantly enhance LLMs' reasoning capabilities Chung et al. (2022); Hsieh et al. (2023); Mukherjee et al. (2023); Fu et al. (2023). Saparov & He (2022) created a synthetic dataset generated based on first-order logic. They then parsed the generated CoT into symbolic proofs for formal analysis. It was shown that LLMs are capable of reasoning. The success of CoT has encouraged researchers to refine the technique and also propose variations of the technique.

For example, Chen et al. (2023) proposed a metric to measure the effectiveness of CoT and a technique to improve CoT for vision-language models. Wang et al. (2023c) studied using multiple reasoning paths and positive and negative answers to improve CoT reasoning. Zhang et al. (2023) proposed cumulative reasoning, which employs LLMs in a cumulative and iterative manner to emulate the

human thought process. Qi et al. (2023) proposed a divide-and-conquer algorithm that simulates the self-questioning and recursive thinking process of humans to improve CoT. Wang & Lu (2023) investigated how to incorporate into relatively small LMs the capabilities of multi-step reasoning and CoT. Wang et al. (2022a) found that even logically invalid CoT also helps to reason. This was confirmed in (Schaeffer et al., 2023). To deal with unsound inferences, Poesia et al. (2023) introduced a class of external tools for LLMs called guides that use states and incremental constraints to guide the generation in reasoning. A related work on using external tools was done in Xu et al. (2023a). Wang et al. (2022b) improved CoT using multiple paths and consistency checks. Ling et al. (2023) studied the verification of CoT. Stolfo et al. (2023) identified part of an LLM responsible for reasoning. In a different direction, Yang et al. (2022) argued that the prevailing approach to CoT prompt selection through trial and error is unsatisfactory. They then proposed a principled approach for multi-domain LLM CoT prompt selection.

Several researchers also broadened the CoT method and proposed the neural symbolic *code prompting* (Hu et al., 2023b), *program of thoughts* (Chen et al., 2022; Cheng et al., 2023), *tree-of-thoughts* (Yao et al., 2023b; Long, 2023), *tree-of-mixed-thoughts* (Hu et al., 2023a), *tree of uncertain thoughts* (Mo & Xin, 2023), *hypergraph-of-thoughts* (Yao et al., 2023a), *recursion of thoughts* (Lee & Kim, 2023), *chain of knowledge* (Wang et al., 2023b), *chain of simultaneous thoughts* (Shao et al., 2022), *graph-of-thoughts* (Yao et al., 2023c), *faithful chain of thoughts* (Lyu et al., 2023), and thought expansion Kim et al. (2023). Further, Bi et al. (2023) proposed a complexity measure and chose the optimal complexity to improve the *program of thoughts* (Chen et al., 2022). Wang et al. (2023a) proposed a method to improve the generation of equations from natural language questions as the intermediate step to answer the original question. Gao et al. (2023) combined CoT and Program-Aided Language Models (PAL) for improved reasoning.

**Empirical work on LG in reasoning.** Many empirical attempts have been made to modify the Transformer and/or learning biases to better solve the LG problem. Duan & Shi (2023) and Zhou et al. (2024) proposed some bias calibration methods to enable the model to learn suitable attention biases. However, their methods are still unable to solve addition perfectly or multiplication at all. Jelassi et al. (2023) proposed to add a small number of long sequences in the training to help solve long sequences, but still could not solve the multiplication problem. Chi et al. (2023) proposed a Transformer variant with weight-sharing, a working memory, etc, to improve LG for regular languages. It can solve some problems but is still unable to deal with multiplication or addition. Different attention and new architectures are also proposed in Nangia & Bowman (2018); Bowman et al. (2015); Tay et al. (2021); Chowdhury & Caragea (2023a;b). However, they don't use CoT but only the direct input and output in training. Their methods work on various text copying and list operations but don't solve these problems and do not work on more complex large-number addition and multiplication. Theoretically, learning to reason without intermediate steps (or CoT) in training is not learnable (Wies et al., 2023; Feng et al., 2023). Our work needs no specialized architectures for different problems, but just a vanilla Transformer with relative position encoding.

# B  PROOFS

In the section, we provide proofs. For ease of reading, we state the same property/theorem again.

For simplicity, denote $\mathcal{H}^c(r)$ as the set of all consistent contexts composed of $r$-length intervals.

For simplicity, denote $\mathcal{H}_n^c(r) = \{h \mid |h| \leq n, h \in \mathcal{H}^c(r)\}$.

**Theorem B.1 (Thm. 3.1, a detailed description)** *Define $V$ to be a metric space. Denote $0 \in V$ to be the empty token. Denote $|v_1 - v_2|$ to be the metric on $V$, $\forall v_1, v_2 \in V$. For $g_N : V^N \to [-1, 1]$, $\forall N' > N$, there exists infinitely many $f_{N'} : V^{N'} \to [-1, 1]$ s.t. (i) $f_{N'}(v_1, \ldots, v_N, 0, \ldots, 0) = g_N(v_1, \ldots, v_N)$, (ii) $f_{N'}$ is Lipchitz-continuous in the neighborhood of $(v_1, \ldots, v_N, 0, \ldots, 0)$ if $g_N$ is Lipchitz-continuous in the neighborhood of $(v_1, \ldots, v_N)$.*

**[Proof.]** Denote $\mathbf{v}_{[1:N]} = (v_1, \ldots, v_N)$. $\forall \epsilon > 0$, let $u : V \to [-1, 1], u(0) = 0$ to be an $\epsilon$-Lipschitz continuous function i.e.

$$|u(v) - u(v')| < \epsilon|v - v'|.$$

There exists infinitely many $u$'s, e.g. $u(v) = \text{sign}(|v - 0|) \cdot \epsilon|v - 0|^\alpha, \ \alpha \leq 1$.

Let

$$f_{N+1}(\mathbf{v}_{[1:N+1]}) = \begin{cases} 1, & u(v_{N+1}) > 1 - g_N(\mathbf{v}_{[1:N]}), \\ -1, & u(v_{N+1}) < -1 - g_N(\mathbf{v}_{[1:N]}), \\ g_N(\mathbf{v}_{[1:N]}) + u(v_{N+1}), & else. \end{cases}$$

Since $u(0) = 0$, we have $f_{N+1}(\mathbf{x}_{[1:N]}, 0) = g_N(\mathbf{x}_{[1:N]})$.

Note that

$$|f_{N+1}(\mathbf{v}_{[1:N+1]}) - f_{N+1}(\mathbf{v}'_{[1:N+1]})|$$
$$\leq |(g_N(\mathbf{v}_{[1:N]}) + u(v_{N+1})) - (g_N(\mathbf{v}'_{[1:N]}) + u(v'_{N+1}))|$$
$$\leq |g_N(\mathbf{v}_{[1:N]}) - g_N(\mathbf{v}'_{[1:N]})| + |u(v_{N+1}) - u(v'_{N+1})|$$
$$\leq |g_N(\mathbf{v}_{[1:N]}) - g_N(\mathbf{v}'_{[1:N]})| + \epsilon|v_{N+1} - v'_{N+1}|.$$

When $g_N$ is Lipschitz continuous in the neighborhood of $\mathbf{v}_{[1:N]}$, we have

$$|g_N(\mathbf{v}_{[1:N]}) - g_N(\mathbf{v}'_{[1:N]})| < \epsilon_{g_N}|\mathbf{v}_{[1:N]} - \mathbf{v}'_{[1:N]}|.$$

Therefore, $|f_{N+1}(\mathbf{v}_{[1:N+1]}) - f_{N+1}(\mathbf{v}'_{[1:N+1]})| \leq (\epsilon_{g_N} + \epsilon)|\mathbf{v}_{[1:N+1]} - \mathbf{v}'_{[1:N+1]}|$. Thus, $f_{N+1}$ is Lipschitz continuous in the neighborhood of $(\mathbf{v}_{[1:N]}, 0)$.

By induction, $\forall N' > N$, there exist infinitely many $f_{N'}$ satisfying the condition (i) and (ii).

**Property B.2 (Prop. 3.4)** *If $h$ is a consistent context and $h \preceq h'$, then $h'$ is also a consistent context.*

**[Proof.]** Denote $\mathscr{S}_h = \{S|\, h \sqsubset S\}$ and $\mathscr{S}_{h'} = \{S|\, h' \sqsubset S\}$. Let $S^0 \in \mathscr{S}_{h'}$, and the CoT step output $S^1$. Since $h \preceq h'$, $h' \sqsubset S^0$ implies $h \sqsubset S^0$, which means $S^0 \in \mathscr{S}_h$. Since $h$ is consistent, we have $S^1[c(h, S^0)] = \psi(h)$. Since $h \preceq h'$ again, we have $c(h, S^0) = c(h', S^0)$, which implies $S^1[c(h', S^0)] = S^1[c(h, S^0)] = \psi(h)$. Therefore, for any $S^0 \in \mathscr{S}_{h'}$, $S^1[c(h', S^0)]$ is always $\psi(h)$, which means that $h'$ is also consistent.

**Property B.3 (Prop. 3.5)** *If $\mathscr{S}$ is $(n, r)$-consistent, then for $\forall n' \geq n$, $\forall r' \geq r$, $\mathscr{S}$ is $(n', r')$-consistent.*

**[Proof.]** Since $\mathscr{S}$ is $(n, r)$-consistent, by definition, for $S^0 \in \mathscr{S}$ and $S^1$ to be the CoT step output, $\forall 1 \leq i \leq |S^1|$, there exists $h_{[i]} \in \mathcal{H}_n^c(r)$ s.t. $c(h_{[i]}, S^0) = i$. Denote $h_{[i]} = (a_1 : a_2, \ldots, a_{|h_{[i]}|})$. since $h_{[i]} \sqsubset S^0$, there exists $1 \leq m_1, \ldots, m_{|h_{[i]}|} \leq |S^0|$ s.t. $s_{m_j} \ldots s_{m_j+r-1} = a_j, 1 \leq j \leq |h_{[i]}|$.

(i) For $\forall n' \geq n$, since $|h_{[i]}| \leq n \leq n'$, which means $h_{[i]} \in \mathcal{H}_{n'}^c(r)$, $\mathscr{S}$ is $(n', r)$-consistent.

(ii) For $\forall r' \geq r$, we construct $h'_{[i]} \in \mathcal{H}_n^c(r')$. Let $left = \lfloor \frac{r'}{2} \rfloor - \lfloor \frac{r}{2} \rfloor$ and $right = \lfloor \frac{r'-1}{2} \rfloor - \lfloor \frac{r-1}{2} \rfloor$. Let $a'_j = s_{m_j-left} \ldots s_{m_j} \ldots s_{m_j+r-1} \ldots s_{m_j+r-1+right}$, $1 \leq j \leq |h_{[i]}|$, then it's not difficult to verify that the central element of $a'_j$ is the central element of $a_j$. Let $h'_{[i]} = (a'_1 : a'_2, \ldots, a'_{|h_{[i]}|})$. If $h'_{[i]}$ is inconsistent, the inconsistent instances containing $h'_{[i]}$ must also contain $h_{[i]}$. However, all central elements of $h'_{[i]}$ and $h_{[i]}$ are the same, which contradicts the fact that $h_{[i]}$ is consistent. Therefore, $h'_{[i]}$ is consistent, i.e. $h'_{[i]} \in \mathcal{H}_n^c(r')$. $\mathscr{S}$ is $(n, r')$-consistent.

Combining (i) and (ii), $\mathscr{S}$ is $(n', r')$-consistent.

**Theorem B.4 (Thm. 3.6)** *If a problem $\mathscr{S}$ is $(n, r)$-consistent, then there exists an $n$-layer Transformer model $f$, for $\forall S^0 \in \mathscr{S}$ with CoT process $(S^0, \ldots, S^T)$, we always have $S^{t+1} = f(S^t), 0 \leq t < T$.*

**[Proof.]** Our key idea is to construct a Transformer model that the predicted CoT step output is free from the length of the input problem instance, which thus achieves LG.

To finish our proof, we borrow 2 lemmas from (Feng et al., 2023). We briefly state them below.

**Lemma B.5 (LOOKUP. Lemma C.5 in (Feng et al., 2023))** *For any lookup table $g : [1, \ldots, d]^k \rightarrow [1, \ldots, d]$, for $\forall \epsilon > 0$, there exists a two-layer MLP with GeLU activation $f : [1, \ldots, d]^k \rightarrow [1, \ldots, d]$ s.t. $|f(x) - g(x)| < \epsilon, \forall x \in [1, \ldots, d]^k$.*

**Lemma B.6 (COPY. Lemma C.7 in (Feng et al., 2023))** *For sequence $x_1, \ldots, x_n$ with scores $r_1, \ldots, r_n$, denote $q_i = W_q x_i, k_i = W_k x_i, v_i = W_v x_i$, for $\forall \epsilon > 0$, $\forall \delta > 0$, the output of a single head attention layer $o_1, \ldots, o_n$ satisfies $|o_i - v_{\arg\max_j \{r_j | q_i \cdot k_j < \delta\}}| < \epsilon, 1 \leq i \leq n$.*

Let's first formulate the structure of $\mathcal{H}_n^c(r)$, which helps apply Lemma B.5 and Lemma B.6 in constructing the Transformer model. Given $h = (a_1 : a_2, \ldots, a_{|h|})$ and an interval/sequence $a$, we denote the operation that concatenates $a$ to the end of $h$ as

$$h \cdot a = (a_1 : a_2, \ldots, a_{|h|}, a).$$

Let $a$ be an arbitrary $r$-length interval. Denote

$$\lambda(a) = \{h | \exists h' \in \mathcal{H}_n^c(r), h \cdot a \preceq h'\}.$$

Note that the context in $\lambda(a)$ needn't be consistent. If $\mathcal{H}_n^c(r)$ only contains contexts that have at most $n$ $r$-length intervals, which is a finite set, then $\lambda(a)$ is also finite. For an arbitrary $a$, we could order $h$'s in $\lambda(a)$ and denote

$$\lambda_o(a) = (h_1, h_2, \ldots, h_{|\lambda(a)|}), \{h | h \in \lambda_o(a)\} = \lambda(a),$$

where $\lambda_o(a)$ represents the *ordered* $\lambda(a)$. Denote

$$N_\lambda = \max_a |\lambda_o(a)|.$$

Let $h$ be an arbitrary context. Denote

$$\beta(h) = \{a | \exists h' \in \mathcal{H}_n^c(r), h \cdot a \preceq h'\}.$$

Similarly, $\beta(h)$ could be *ordered* for each $h$, which is

$$\beta_o(h) = (a_1, a_2, \ldots, a_{|\beta(h)|}), \{a | a \in \beta_o(h)\} = \beta(h).$$

Denote

$$N_\beta = \max_h |\beta_o(h)|.$$

**Transformer Construction**

Now we construct a Transformer model. Denote the input problem instance as $S^0 = s_1 \ldots s_{|S^0|}$ and the CoT step output as $S^1$. Since $\mathscr{S}$ is $(n, r)$-consistent, for $1 \leq i \leq |S^1|$, there exists $h_{[i]} \in \mathcal{H}_n^c(r)$ s.t. $c(h_{[i]}, S^0) = i$ and $S^1[i] = \psi(h_{[i]})$.

For simplicity, we denote the $r$-length interval that centers at $i$ as

$$a_{[i]} = s_{i - \lfloor \frac{r}{2} \rfloor} \ldots s_i \ldots s_{i + \lfloor \frac{r-1}{2} \rfloor}.$$

**Layer 1.** The 1st attention layer applies a padding mask with relative position encoding. For each $i$, the padding mask only exposes the $r$-length interval $a_{[i]}$. Denote

$$h_{[i]} = (a_{[i]}).$$

Then, after the 1st attention layer, each vector at position index $i$ contains

$$e_i = (a_{[i]}, h_{[i]}).$$

We further make $N_\beta$ copies of $h_{[i]}$ and write

$$p_{[i]} = (h_{[i]}, \ldots, h_{[i]}), |p_{[i]}| = N_\beta.$$

Then, each vector at position index $i$ contains

$$e_i = (a_{[i]}, p_{[i]}),$$

where $p_{[i]}$ contains $N_\beta$ *potential* contexts, and we use $p_{[i],j}$ to represent the $j$-th context of $p_{[i]}$.

The 1st feed-forward (FFD) layer maps $a_{[i]}$ to $\lambda_o(a_{[i]})$. Since $\lambda_o(a_{[i]}) = (h_1, \ldots, h_{|\lambda(a_{[i]})|})$, it induces a lookup table

$$(a_{[i]}, j) \to h_j, 1 \leq j \leq |\lambda(a_{[i]})| \leq N_\lambda,$$

where $h_j$ is represented by any form that could be distinguished from other $h$'s (e.g. one-hot for $|h| \leq n$). Therefore, by Lemma B.5, we could construct at most $N_\lambda$ lookup tables that maps $a_{[i]}$ to $\lambda_o(a_{[i]})$. Then after the 1st FFD layer, each vector at position index $i$ contains

$$e_i = (a_{[i]}, \lambda_o(a_{[i]}), p_{[i]}).$$

Note that $\lambda_o(a_{[i]})$ contains $N_\lambda$ contexts, and we use $\lambda_o(a_{[i]})_j$ to represent the $j$-th context of $\lambda_o(a_{[i]})$.

**Layer 2 to Layer $n$.** The following layers share the same procedure. We describe the construction of the 2nd layer in detail, and the remaining layers are the same.

The 2nd attention layer has $N_\beta \times N_\lambda$ attention heads. For the $(j_1, j_2)$-th head, let $W_q, W_k, W_v$ be

$$q_i = W_k e_i = p_{[i], j_1}, k_i = W_q e_i = \lambda_o(a_{[i]})_{j_2}, v_i = W_v e_i = a_{[i]}.$$

Then it's not difficult to find that in Lemma B.6, the *matching set* $\{q_{i_1} \cdot k_{i_2} < \delta\}$ has at most 1 element only when $p_{[i_1], j_1} = \lambda_o(a_{[i_2]})_{j_2}$. By Lemma B.6, the $(j_1, j_2)$-th head's output for position index $i$ is

$$o_i = \begin{cases} p_{[i], j_1} \cdot a_{[i']}, & p_{[i], j_1} = \lambda_o(a_{[i']})_{j_2}, \\ p_{[i], j_1}, & otherwise. \end{cases}$$

Among these $N_\beta \times N_\lambda$ heads, by definition of $N_\beta$, there exist at most $N_\beta$ heads that concatenate a new interval. Denote the output of these heads as $p_{[i]}$ again. Then, after the 2nd attention layer, $p_{[i]}$ is updated, and each vector of position index $i$ is

$$e_i = (a_{[i]}, \lambda_o(a_{[i]}), p_{[i]}).$$

The 2nd FFD layer verifies whether one of $p_{[i]}$ has already been a consistent context. For any $|h| \leq n$, $h$ is either in $\mathcal{H}_n^c(r)$ or not, which induces a lookup table $h \to \{0, 1\}$. Therefore, by Lemma B.5, there exists $N_\beta$ FFD layers to verify whether each context of $p_{[i]}$ has already been consistent. Then, after the 2nd FFD layer, each vector of position index $i$ is

$$e_i = (a_{[i]}, \lambda_o(a_{[i]}), p_{[i]}, \mathbf{1}_{[i]}),$$

where $\mathbf{1}_{[i]}$ has $N_\beta$ boolean, and the $j$-th boolean indicates whether $p_{[i], j}$ is consistent.

For the remaining Transformer layers, the attention layer updates $p_{[i]}$ and the FFD layer updates $\mathbf{1}_{[i]}$.

**Output** After $n$ Transformer layers, since $\mathscr{S}$ is $(n, r)$-consistent, for each position index $i$, there must exist at least one consistent context $h$ in $p_{[i]}$, i.e. at least one positive boolean in $\mathbf{1}_{[i]}$. Since $h \to \psi(h)$ induces a lookup table, by Lemma B.5, we simply map the consistent context $h$ into $\psi(h)$ by a FFD layer. When $h$ is consistent and $c(h, S^0) = i$, by definition, $\psi(h)$ is exactly $S^1[i]$.

## C   A METHOD TO ACHIEVE $(n, r)$-CONSISTENCY

We first introduce a measure induced by Def. 3.3 that describes how far a problem is from $(n, r)$-consistency. Let $h = (a_1 : a_2, \ldots, a_{|h|})$ be a context. For problem instances $\leq N$, let $\mu_N$ be a measure of $h$'s.[10] Let $p(\psi(h)|h)$ be the probability density of $\psi(h)$ conditioned on $h$, where $\psi(h)$ is the central element of the next CoT step of the anchor interval of $h$ (Def. 3.3 (v)). Denote $\mathcal{E}$ as the entropy function and $\mathcal{E}(p(\psi(h)|h))$ the entropy of $p(\psi(h)|h)$. Define

$$\Omega(N; n, r) = \int \mathcal{E}(p(\psi(h)|h)) \mu_N(h) dh. \tag{1}$$

Note that when a problem is $(n, r)$-consistent, by definition, $\forall h$, $p(\psi(h)|h)$ is a Dirac delta function. Therefore, $\mathcal{E}(p(\psi(h)|h)) = 0$ and $\Omega(N; n, r) = 0$. It's not difficult to see that a problem is $(n, r)$-consistent *if and only if* $\limsup_{N \to +\infty} \Omega(N; n, r) = 0$.

Therefore, when a problem is not $(n, r)$-consistent, i.e,. $\Omega(N; n, r) > 0$, there exists a method to transform it into $(n, r)$-consistent. When $\Omega(N; n, r) > 0$, there exists $h$ s.t. $\mathcal{E}(p(\psi(h)|h)) > 0$. For

---

[10]E.g. let $u$ to be a uniform measure over all problem instances $\leq N$ and let $v$ to be a uniform measure of all context given a problem instance $S$. Then $\mu_N(h) = \int v(h|S)\mu(S)dS$ defines a measure of $h$'s.

this $h$, let $(A, B)$ to be a separation of the vocabulary, i.e. $A \cup B = V$ and $A \cap B = \emptyset$. It's obvious that

$$\mathcal{E}(p(\psi(h)|h, \psi(h) \in A)) + \mathcal{E}(p(\psi(h)|h, \psi(h) \in B)) \leq \mathcal{E}(p(\psi(h)|h)), \tag{2}$$

where $=$ holds iff $\{\psi(h)|h\} \cap A = \emptyset$ or $\{\psi(h)|h\} \cap B = \emptyset$. Therefore, whenever $|\{\psi(h)\}| > 1$, there always exists a separation s.t. $\mathcal{E}(p(\psi(h)|h))$ decreases strictly. Therefore, we could add a tag (A, h) for $\psi(h) \in A$ and a tag (B, h) for $\psi(h) \in B$. Repeating adding tags for $h$ will finally decrease $\mathcal{E}(p(\psi(h)|h))$ to 0. Repeating dealing with all possible $h$ with $\mathcal{E}(p(\psi(h)|h)) > 0$ will finally decrease $\Omega(N; n, r)$ to 0.

This method adds different tags for different $h$'s. This is inefficient for most problems, as many tags could have been shared among $h$'s. Listing all tags for all $h$'s without reduction makes this method computationally intractable. It's unknown whether there exists a more computationally efficient method to transform a problem into $(n, r)$-consistent.

## D  SUPPLEMENTARY EXPERIMENTS

In addition to the main results reported in Fig. 1, we add more tests with longer lengths to stress test the system. Three more test settings are added, i.e. LG Test 6, 7, and 8. The full set of test settings and their corresponding results are given in Table 2 and Table 3 respectively.

Table 2: Experimental settings. The settings from Train Length to LG Test 5 have already been given in Table 1. LG Test 6, 7, and 8 are additional settings with longer lengths.

| | Train Length | LG Test 1 | LG Test 2 | LG Test 3 | LG Test 4 | LG Test 5 | LG Test 6 | LG Test 7 | LG Test 8 |
|---|---|---|---|---|---|---|---|---|---|
| arithmetic in $F_7$ | $L \in [3, 20)$ | $L \in [3, 30)$ | $L \in [3, 40)$ | $L \in [3, 50)$ | $L \in [3, 60)$ | $L \in [3, 100)$ | $L \in [3, 200)$ | $L \in [3, 500)$ | $L \in [3, 1000)$ |
| parity-[2] | $L \in [1, 8)$ | $L \in [1, 30)$ | $L \in [1, 40)$ | $L \in [1, 50)$ | $L \in [1, 60)$ | $L \in [1, 100)$ | $L \in [1, 200)$ | $L \in [1, 500)$ | $L \in [1, 1000)$ |
| addition-[2] | $L \in [1, 8)$ | $L \in [1, 9)$ | $L \in [1, 10)$ | $L \in [1, 11)$ | $L \in [1, 16)$ | $L \in [1, 21)$ | $L \in [1, 31)$ | $L \in [1, 41)$ | $L \in [1, 51)$ |
| multiplication-[11] | $L \in [1, 6)$ | $L \in [1, 7)$ | $L \in [1, 8)$ | $L \in [1, 9)$ | $L \in [1, 10)$ | $L \in [1, 11)$ | $L \in [1, 16)$ | $L \in [1, 21)$ | $L \in [1, 26)$ |
| division-[12] | $L \in [1, 6)$ | $L \in [1, 7)$ | $L \in [1, 8)$ | $L \in [1, 9)$ | $L \in [1, 10)$ | $L \in [1, 11)$ | $L \in [1, 16)$ | $L \in [1, 21)$ | $L \in [1, 26)$ |

Table 3: Experimental results. The results up to LG Test 5 have been shown in Fig. 1. Notice that LG Test 5 is 100%, but LG Test 2, 3, and 4 are not. This is possible because each experiment randomly generates its training and test data.

| Accuracy (%) | Train Length | LG Test 1 | LG Test 2 | LG Test 3 | LG Test 4 | LG Test 5 | LG Test 6 | LG Test 7 | LG Test 8 |
|---|---|---|---|---|---|---|---|---|---|
| arithmetic in $F_7$ | 100.0 | 100.0 | 100.0 | 100.0 | 100.0 | 100.0 | 99.0 | 96.4 | 91.8 |
| parity-[2] | 100.0 | 100.0 | 100.0 | 100.0 | 100.0 | 100.0 | 100.0 | 100.0 | 100.0 |
| addition-[2] | 100.0 | 100.0 | 100.0 | 100.0 | 100.0 | 100.0 | 100.0 | 90.0 | 77.5 |
| multiplication-[11] | 100.0 | 100.0 | 100.0 | 100.0 | 100.0 | 100.0 | 98.9 | 70.0 | 61.4 |
| division-[12] | 100.0 | 100.0 | 99.8 | 99.6 | 99.6 | 100.0 | 98.8 | 81.4 | 65.7 |

We can observe that although these problems are $(n, r)$-consistent, they still struggle to achieve perfect LG for extremely long lengths. The key reason is due to the "dense" property of the attention layer, which involves noise beyond the desired $n$ intervals as the length increases. Fig. 2 gives an example to show the dilution of the probability distribution.

## E  COMPUTING RESOURCES

Each experiment is running on a machine with 8 CPU cores. Each experiment takes less than 24 hours.

## F  CoT EXAMPLES

Fig. 3 shows some examples of CoT schemes of the experimental reasoning problems. Note that the **parity** problem uses '?' in the second line of the input and output to represent the position to be calculated next, 1 to represent *odd*, and 0 to represent *even*. The first line in the input is the input bit sequence. The **addition** problems use '?' to represent 0 being carried from the right and '$' to represent 1 being carried from the right. In all problems, * is equivalent to ×. We use * instead of × for ease of aligning chars. The orange lines in the output, which are for easy reading, are not predicted in practice, as they are identical to lines in the input.

Table 4 lists all the indicator tokens used in *multiplication*-[11]. Table 5 lists all the indicator tokens used in *division*-[12].

Figure 2: An example of a diluted probability distribution. We present the attention probability distribution on the first CoT step of $123 \times 789/123456789/123456789123456789$, where the query is 3 and the key/value is $789/123456789/123456789123456789$, respectively. The increased number of tokens takes attention away from the desired tokens, which degrades the performance.

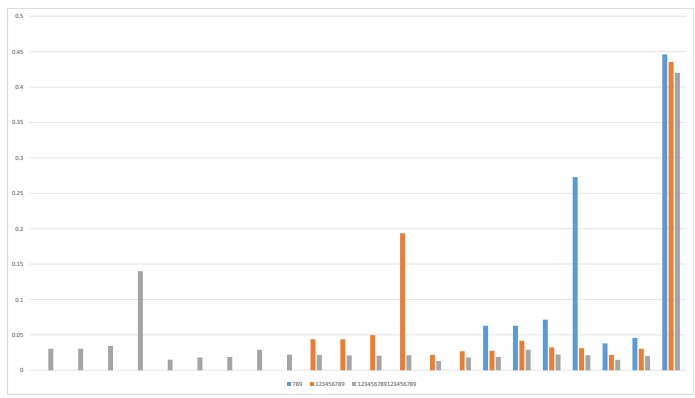

Table 4: Descriptions of indicator tokens for *multiplication*-[11].

| Indicator | Dynamic | Description |
|---|---|---|
| E, S | do not move | start and end of $a$ |
| e, s | do not move | start and end of $b$ |
| I | move left when J moves back to s | digit of $a$ to multiply |
| J | move left; when J reaches e, J moves back to s | digit of $b$ to multiply |
| F, T | move left when J moves back to s | start and end of K; T is aligned with I, |
|  |  | distance between F and T is the same as distance between e and s |
| K | move left; when K reaches F, K moves back to T | indicator of output position index |
| ?, c | move left; ? appears after multiplication | ? carries 0 and c carries 1 in addition |
| # | appear when addition is done | # indicates next CoT step to be multiplication |

Table 5: Descriptions of indicator tokens for *division*-[12].

| Indicator | Dynamic | Description |
|---|---|---|
| E, S | do not move | start and end of $c$ ($c \div a = b$) |
| e, s | do not move | start and end of $a$ |
| I | move right when J moves back to s | digit of $a$ to subtract |
| J | move left; when J reaches e or subtraction fails, J moves back to s | digit of $b$ to be subtract |
| F, T | move right when J moves back to s | start and end of K; T is aligned with I, |
|  |  | distance between F and T is the same as distance between e and s |
| K | move left; when K reaches F or subtraction fails, K moves back to T | indicator of output position index |
| b | appear when subtraction needs to borrow 1 | b borrows 1 in subtraction |

Figure 3: Examples of the CoT schemes in Sec. 4. Note that the figure covers *multiple pages*. **multiplication-[11]** and **division-[12]** appear in the next three pages.

**(a) arithmetic in prime field $F_7$** (note, this is not the normal arithmetic)

| | |
|---|---|
| Input[0]: | (0 + 4  - ( 2 - 3 * 6)) * (4 + 0) |
| Output[0]: | (  4    - ( 2 -  4  )) *    4 |
| Input[1]: | (4 - (2 - 4)) * 4 |
| Output[1]: | (4 -   5   ) * 4 |
| Input[2]: | (4 - 5) * 4 |
| Output[2]: |    6    * 4 |
| Input[3]: | 6 * 4 |
| Output[3]: |    3 |

**(e) multiplication-[1]**

| | |
|---|---|
| Input[0]: | 1 * 3 = ? |
| Output[0]: | 1 * 3 = 1 + ? |
| Input[1]: | 1 * 3 = 1 + ? |
| Output[1]: | 1 * 3 = 1 + 1 + ? |
| Input[2]: | 1 * 3 = 1 + 1 + ? |
| Output[2]: | 1 * 3 = 1 + 1 + 1 |
| Input[3]: | 1 * 3 = 1 + 1 + 1 |
| Output[3]: | 1 * 3 = 2 + 1 |
| Input[4]: | 1 * 3 = 2 + 1 |
| Output[4]: | 1 * 3 = 3 |

**(b) parity-[2]**

| | |
|---|---|
| Input[0]: | 1011 ? |
| Output[0]: | 1011 1? |
| Input[1]: | 1011 1? |
| Output[1]: | 1011 11? |
| Input[2]: | 1011 11? |
| Output[2]: | 1011 110? |
| Input[3]: | 1011 110? |
| Output[3]: | 1011 1011 |

**(c) addition-[1]**

| | |
|---|---|
| Input[0]: | 285+9805= ? |
| Output[0]: | 285+9805=$0 |
| Input[1]: | 285+9805= $0 |
| Output[1]: | 285+9805=?90 |
| Input[2]: | 285+9805= ?90 |
| Output[2]: | 285+9805=$090 |
| Input[3]: | 285+9805= c090 |
| Output[3]: | 285+9805=10090 |

**(d) addition-[2]**

| | |
|---|---|
| Input[0]: | 285 +  9805 =   ?
I        J    K |
| Output[0]: | 285 +  9805 = $0
I        J    K |
| Input[1]: | 285 +  9805 =   $0
I        J    K |
| Output[1]: | 285 +  9805 = ?90
I        J    K |
| Input[2]: | 285 +  9805 =   ?90
I        J    K |
| Output[2]: | 285 +  9805 = $090
I        J    K |
| Input[3]: | 285 +  9805 =   c090
I        J    K |
| Output[3]: | 285 +  9805 = 10090
I    J    K |

**(f) multiplication-[11]**

| Input[0]: | 1 2 | Input[2]: | 1 2 | Input[4]: | 1 2 | Input[6]: | 1 2 | Input[8]: | 1 2 |
|---|---|---|---|---|---|---|---|---|---|
| | E S | | E S | | E S | | E S | | E S |
| | I | | I | | I | | I | | I |
| | 3 4 | | 3 4 | | 3 4 | | 3 4 | | 3 4 |
| | e s | | e s | | e s | | e s | | e s |
| | J | | J | | J | | J | | i |
| | 8 | | 8 | | 6 8 | | 0 8 | | 1 0 8 |
| | F T | | F T | | F T | | F T | | F T |
| | K | | K | | K | | K | | K |
| | | | | | | | | | 3 |
| | # | | # | | # | | c | | ? |
| Output[0]: | 1 2 | Output[2]: | 1 2 | Output[4]: | 1 2 | Output[6]: | 1 2 | Output[8]: | 1 2 |
| | E S | | E S | | E S | | E S | | E S |
| | I | | I | | I | | I | | I |
| | 3 4 | | 3 4 | | 3 4 | | 3 4 | | 3 4 |
| | e s | | e s | | e s | | e s | | e s |
| | J | | J | | J | | J | | J |
| | F T | | F T | | F T | | F T | | F T |
| | K | | K | | K | | K | | K |
| | 8 | | 6 | | 4 | | # | | # |
| | ? | | ? | | ? | | 1 0 8 | | 4 0 8 |
| | | | 8 | | 6 8 | | | | |

| Input[1]: | 1 2 | Input[3]: | 1 2 | Input[5]: | 1 2 | Input[7]: | 1 2 |
|---|---|---|---|---|---|---|---|
| | E S | | E S | | E S | | E S |
| | I | | I | | I | | I |
| | 3 4 | | 3 4 | | 3 4 | | 3 4 |
| | e s | | e s | | e s | | e s |
| | J | | J | | J | | J |
| | 8 | | 8 | | 6 8 | | 1 0 8 |
| | F T | | F T | | F T | | F T |
| | K | | K | | K | | K |
| | 8 | | 6 | | 4 | | # |
| | ? | | ? | | ? | | |
| Output[1]: | 1 2 | Output[3]: | 1 2 | Output[5]: | 1 2 | Output[7]: | 1 2 |
| | E S | | E S | | E S | | E S |
| | I | | I | | I | | I |
| | 3 4 | | 3 4 | | 3 4 | | 3 4 |
| | e s | | e s | | e s | | e s |
| | J | | J | | J | | J |
| | F T | | F T | | F T | | F T |
| | K | | K | | K | | K |
| | | | | | | | | 3 |
| | # | | # | | c | | ? |
| | 8 | | 6 8 | | 0 8 | | 1 0 8 |

**(g) division-[12] (i)**

| Input[0]: | | 4 | 0 | 8 | | Input[2]: | | 4 | 0 | 8 | | Input[4]: | | 4 | 0 | 8 | | Input[6]: | | 4 | 0 | 8 |
|---|---|---|---|---|---|---|---|---|---|---|---|---|---|---|---|---|---|---|---|---|---|---|---|
| | E | | | S | | | E | | | S | | | E | | | S | | | E | | | S |
| 3 | 4 | | | | | 3 | 4 | | | | | 3 | 4 | | | | | 3 | 4 | | | | |
| e | s | | | | | e | s | | | | | e | s | | | | | e | s | | | | |
| | | 4 | 0 | 8 | | | | 4 | 0 | 8 | | | | 4 | 0 | 8 | | | | | 6 | 8 |
| | | 4 | 0 | 8 | | | | 4 | 0 | 8 | | | | | 6 | 8 | | | | | 2 | 8 |
| | | I | | | | | | I | | | | | | I | | | | | | I | | |
| | J | | | | | | J | | | | | | J | | | | | | J | | | |
| | F | | T | | | | F | | T | | | | F | | T | | | | F | | T | |
| | K | | | | | | | K | | | | | K | | | | | | K | | | |
| | | 0 | 0 | 0 | | | | 0 | 0 | 0 | | | | 0 | 0 | 0 | | | | 0 | 1 | 0 |
| Output[0]: | | I | | | | Output[2]: | | I | | | | Output[4]: | | I | | | | Output[6]: | | I | | |
| | J | | | | | | J | | | | | | J | | | | | | J | | | |
| | F | | T | | | | F | | T | | | | F | | T | | | | F | | T | |
| | K | | | | | | | K | | | | | K | | | | | | | K | |
| | | 4 | 0 | 8 | | | | 4 | 0 | 8 | | | | | 6 | 8 | | | | | 6 | 8 |
| | | | | 8 | | | | 4 | 6 | 8 | | | | | 6 | 8 | | | | | 6 | 8 |
| | | | | | | | | b | | | | | | | | | | | | | | |
| | | 0 | 0 | 0 | | | | 0 | 0 | 0 | | | | 0 | 1 | 0 | | | | 0 | 1 | 0 |

| Input[1]: | | 4 | 0 | 8 | | Input[3]: | | 4 | 0 | 8 | | Input[5]: | | 4 | 0 | 8 | | Input[7]: | | 4 | 0 | 8 |
|---|---|---|---|---|---|---|---|---|---|---|---|---|---|---|---|---|---|---|---|---|---|---|---|
| | E | | | S | | | E | | | S | | | E | | | S | | | E | | | S |
| 3 | 4 | | | | | 3 | 4 | | | | | 3 | 4 | | | | | 3 | 4 | | | | |
| e | s | | | | | e | s | | | | | e | s | | | | | e | s | | | | |
| | | 4 | 0 | 8 | | | | 4 | 0 | 8 | | | | | 6 | 8 | | | | | 6 | 8 |
| | | | | 8 | | | | 4 | 6 | 8 | | | | | 6 | 8 | | | | | 6 | 8 |
| | | I | | | | | | I | | | | | | I | | | | | | | I | |
| | J | | | | | | J | | | | | | J | | | | | | J | | | |
| | F | | T | | | | F | | T | | | | F | | T | | | | F | | T | |
| | K | | | | | | | K | | | | | K | | | | | | | K | |
| | | | | | | | | b | | | | | | | | | | | | | | |
| | | 0 | 0 | 0 | | | | 0 | 0 | 0 | | | | 0 | 1 | 0 | | | | 0 | 1 | 0 |
| Output[1]: | | I | | | | Output[3]: | | I | | | | Output[5]: | | I | | | | Output[7]: | | I | | |
| | J | | | | | | J | | | | | | J | | | | | | J | | | |
| | F | | T | | | | F | | T | | | | F | | T | | | | F | | T | |
| | K | | | | | | | K | | | | | K | | | | | | | K | |
| | | 4 | 0 | 8 | | | | 4 | 0 | 8 | | | | | 6 | 8 | | | | | 6 | 8 |
| | | 4 | 0 | 8 | | | | | 6 | 8 | | | | | 2 | 8 | | | | | 6 | 4 |
| | | 0 | 0 | 0 | | | | 0 | 0 | 0 | | | | 0 | 1 | 0 | | | | 0 | 1 | 0 |

**(g) division-[12] (ii)**

| Input[8]: | | 4 | 0 | 8 |
|---|---|---|---|---|
| | E | | | S |
| 3 | 4 | | | |
| e | s | | | |
| | | 6 | | 8 |
| | | 6 | | 4 |
| | | | | I |
| J | | | | |
| | F | | | T |
| | K | | | |
| | 0 | 1 | | 0 |

| Output[8]: | | | | I |
|---|---|---|---|---|
| J | | | | |
| | F | | | T |
| | K | | | |
| | | 6 | | 8 |
| | | 3 | | 4 |
| | 0 | 1 | | 0 |

| Input[10]: | | 4 | 0 | 8 |
|---|---|---|---|---|
| | E | | | S |
| 3 | 4 | | | |
| e | s | | | |
| | | 3 | | 4 |
| | | 3 | | 4 |
| | | | | I |
| J | | | | |
| | F | | | T |
| | | | | K |
| | 0 | 1 | | 1 |

| Output[10]: | | | | I |
|---|---|---|---|---|
| J | | | | |
| | F | | | T |
| | | | | K |
| | | 3 | | 4 |
| | | 3 | | 0 |
| | 0 | 1 | | 1 |

| Input[12]: | | 4 | 0 | 8 |
|---|---|---|---|---|
| | E | | | S |
| 3 | 4 | | | |
| e | s | | | |
| | | 3 | | 4 |
| | | | | 0 |
| | | | | I |
| J | | | | |
| | F | | | T |
| | K | | | |
| | 0 | 1 | | 1 |

| Output[12]: | | | | I |
|---|---|---|---|---|
| J | | | | |
| | F | | | T |
| | K | | | |
| | | | | 0 |
| | | | | 0 |
| | 0 | 1 | | 2 |

| Input[14]: | | 4 | 0 | 8 |
|---|---|---|---|---|
| | E | | | S |
| 3 | 4 | | | |
| e | s | | | |
| | | | | 0 |
| | | | | 6 |
| | | | | I |
| J | | | | |
| | F | | | T |
| | K | | | |
| | | | | b |
| | 0 | 1 | | 2 |

| Output[14]: | | | | |
|---|---|---|---|---|
| J | | | | |
| | F | | | |
| | | | | 0 |
| | | | | 0 |
| | 0 | 1 | | 2 |

| Input[9]: | | 4 | 0 | 8 |
|---|---|---|---|---|
| | E | | | S |
| 3 | 4 | | | |
| e | s | | | |
| | | 6 | | 8 |
| | | 3 | | 4 |
| | | | | I |
| J | | | | |
| | F | | | T |
| | K | | | |
| | 0 | 1 | | 0 |

| Output[9]: | | | | I |
|---|---|---|---|---|
| J | | | | |
| | F | | | T |
| | | | | K |
| | | 3 | | 4 |
| | | 3 | | 4 |
| | 0 | 1 | | 1 |

| Input[11]: | | 4 | 0 | 8 |
|---|---|---|---|---|
| | E | | | S |
| 3 | 4 | | | |
| e | s | | | |
| | | 3 | | 4 |
| | | 3 | | 0 |
| | | | | I |
| J | | | | |
| | F | | | T |
| | K | | | |
| | 0 | 1 | | 1 |

| Output[11]: | | | | I |
|---|---|---|---|---|
| J | | | | |
| | F | | | T |
| | K | | | |
| | | 3 | | 4 |
| | | | | 0 |
| | 0 | 1 | | 1 |

| Input[13]: | | 4 | 0 | 8 |
|---|---|---|---|---|
| | E | | | S |
| 3 | 4 | | | |
| e | s | | | |
| | | | | 0 |
| | | | | 0 |
| | | | | I |
| J | | | | |
| | F | | | T |
| | | | | K |
| | 0 | 1 | | 2 |

| Output[13]: | | | | I |
|---|---|---|---|---|
| J | | | | |
| | F | | | T |
| | | | | K |
| | | | | 0 |
| | | | | 6 |
| | | | | b |
| | 0 | 1 | | 2 |

