# OpenReview forum: "Generalizing Reasoning Problems to Longer Lengths"
_ICLR.cc/2025/Conference — ICLR 2025 Poster_

### Official Review · Reviewer_AyZa · 2024-11-03

**Soundness:** 3
**Presentation:** 3
**Contribution:** 3
**Rating:** 8
**Confidence:** 4

**Summary:**

This paper addresses the problem of generalizing chain-of-thought (CoT) reasoning to arbitrary lengths, which is called Length Generalization (LG). The previous work has only shown the effective generalization of CoT over inputs of the same length.
The paper first gives a theorem characterizing the root cause of the LG problem - namely there exist infinitely many Lipschitz-continuous extensions of a Lipschitz-continuous function g over N inputs. This implies that LG cannot be achieved in general without some restrictions. It then proves a condition called (n-r) consistency under which LG can be achieved. The paper gives experimental results that show that algorithms for parity, addition, multiplication, and division are learned by transformers with CoT representations that satisfy (n-r)-consistency.

**Strengths:**

LG is an important problem that demonstrates the power of transformers to learn procedures over inputs of arbitrary size.

The paper presents a sound theory that characterizes the root cause of the problem and gives a sufficient condition for its solution.

The experimental results are compelling and illustrate the successes and failures of generalization that depend on the (n,r)-consistency.

**Weaknesses:**

The CoT training seems to be quite tedious for complex procedures such as multiplication.

Typo:Def 3. ... are r-length.

**Questions:**

How robust is the approach to errors in the steps of the CoT? Can you conduct some experiments to study this and report in the final paper?

---

> ### Author Response · Authors · 2024-11-22
> **Response 1 to Reviewer AyZa**
>
> $\textbf{Dear Reviewer AyZa,}$
>
> Thank you very much for your positive comments. We hope this response can address your question.
>
> $\textbf{Q1}$: How robust is the approach to errors in the steps of the CoT? Can you conduct some experiments to study this and report in the final paper?
>
> $\textbf{Response}$: We have experimented on parity-[2] by choosing a token at random in each CoT step and replacing it with another token chosen at random from the vocabulary, and found that its accuracy is still 1.0 on the tested lengths. We'll continue adding more noise on parity and other problems and report the results in the final revised paper.

---

### Official Review · Reviewer_sCy1 · 2024-11-04

**Soundness:** 3
**Presentation:** 3
**Contribution:** 2
**Rating:** 6
**Confidence:** 4

**Summary:**

**Update after rebuttal:** While some of my criticism remains, the authors have clarified some misunderstandings and questions I have had, and have significantly sharpened the focus of the paper (defining a problem class for which LG can be theoretically guaranteed; as well as putting the emphasis on the empirical side). Accordingly, I have raised my score from a 5 to a 6.

---

This paper investigates the challenge of length generalization with transformers. The paper first points out the well-known fact that the induction problem has no correct solution. For length generalization, additional assumptions are thus necessary. The paper proposes to restrict length generalization to a class of problems that can be decomposed into a set of (n) repeated subproblems that only require local context (of length r) around a set of anchor positions: (n,r) consistency. This “necessary” condition (which is a bit of a tautology because the problem class was defined according to the necessary condition) is then shown to be sufficient for the problem to be solved with transformers in theory (ignoring questions around learnability and model capacity, etc.). As the paper notes in 3.5, there is no general method to implement the process that decomposes an original length generalization problem into a “Chain-of-Thought” (CoT) formulation that can be solved by sequential execution of a number of local subproblems. But the paper correctly argues that if solving this problem reformulation such that (n,r) consistency can be ensured, transformers should suffice in principle to solve the modified problem. (n,r) consistency can thus be seen as a guiding target for CoT formulations. A small set of empirical results confirms the relevance of the theoretical result in practice. For training the CoT decomposition of the original problem is done by manually designed, problem specific algorithms, and transformers are trained to execute these algorithms of stepwise rewriting of the problem and simultaneously solving suitable subproblems during rewriting.

**Strengths:**

* Timely and important problem: length generalization even on simple tasks is a regime where transformers struggle.
* Definition of an interesting class of length generalization problems: the class of problems where a decomposition with (n,r) consistency is possible. These problems match well onto, and are probably solvable (in theory) by, transformers.
* Good empirical results on historically challenging problems like parity, addition, multiplication and division.

**Weaknesses:**

* The CoT decomposition does most of the heavy lifting - but it is done manually. The paper makes very little progress here, except defining (n,r) consistency as a guiding principle to aim for (but with no systematic process to get there, and no theoretical understanding of how the class of (n,r) consistent decomposable problems relates to general function and complexity classes (e.g., regular languages).
* Writing throughout most of the manuscript makes claims that are either too grand or not sharp enough, which makes the first part of the paper a bit misleading. For example, (n,r) consistency is not *necessary* to solve length generalization in general. There are loads of other approaches that have been proposed before and that are theoretically very well understood - they typically use different biases, most commonly low-complexity biases, to answer the induction problem (such as Solomonoff Induction or Bayesian inference for example). What is correct is that when restricting to problems that allow for a (n,r) consistent decomposition, length generalization can be solved uniquely without requiring additional biases. I will be more specific in the Questions section.
* Learnability and generalization with transformers in practice remain unclear (bounded capacity, finite data, SGD). The CoT decomposition works on the simple examples shown, but it is not clear whether this would hold at scale (the number of (n,r) consistent CoT steps may scale very badly for some problems); or even how to do this - e.g., how would one train a transformer that can solve all tasks in the paper simultaneously. How would one go about eventually reaching the scale of modern frontier models?

**Verdict**
Overall I think the research in the paper is on a good track, but the current theoretical understanding and presentation is lacking a bit. Large parts of the paper sound like the paper makes a fundamental theoretical contribution that solves length generalization once and for all (such as “a theorem to show the LG problem’s root cause”, or a ”necessary [theoretical condition] to resolve it.” ). Actually the “root cause” (theorem (3.1)) simply points out the very well known and much discussed fact that the induction problem (of which length generalization is one instance) has no correct solution (unlike logical deduction) only plausible or highly likely continuations, and that additional inductive biases are needed (commonly used are low-complexity biases such as Occam’s razor, formalised in many different ways). The “necessary” condition introduced in Sec. 3 is nothing but a restriction to a subset of length generalization problems (the condition is not necessary in general, it simply defines a problem class for which a solution strategy and uniqueness of the solution can be proven). I would strongly prefer if the paper were rewritten to acknowledge this (from the very beginning and throughout), by saying: here is a formal definition of a problem class for which length generalization with transformers can be proved theoretically, and empirical results align well with the theory. For a theory paper I would then like to see more theoretical understanding of this problem class; which length generalization problems can be massaged to fall into this class, and what can be said about the process to do this (can we bound the number of CoT steps in a meaningful way, how does the problem class relate to other well known complexity classes, etc). Alternatively the theory can be left lightweight as is, and the empirical part could be strengthened - but since the CoT decompositions must be manually designed, this is quite tedious. Overall I think the work is ready for presentation and discussion (and notions similar to (n,r) consistency have popped up recently in other works regarding length generalization), but the best format currently, I think, is a workshop. This is not meant to discourage the authors (after all the empirical results on addition, parity, and multiplication are nice - but they are mainly due to handcrafting), but I think a much stronger version of the manuscript is possible with a bit more time and a major revision.

**Questions:**

**Improvements** (that would make me raise my score, but may not be possible within the short rebuttal time)

1. A significant overhaul / rewrite to emphasize that the main contribution is not a theory of length generalization and identification of a novel root cause (the root cause is well known in machine learning, mathematics and philosophy), but a definition of a problem class that has favorable properties w.r.t. length generalization built in, and these properties can be exploited by transformers.
2. If the focus of a major revision lies on the theory (though I personally would recommend leaning more towards the empirical side), then the problem class of (n,r) consistent problems needs to be better understood theoretically. E.g., how does it relate to commonly known complexity classes such as regular languages or $TC^0$? Is this class likely to be complete, i.e., all problems where transformers can length generalize must be (n,r) consistent (which I personally doubt), and if not, what are counterexamples?
3. Strengthen the point of how the current theoretical understanding can help design learnable CoT schemes that generalize. The empirical examples in the paper are good, but how much did the theoretical understanding contribute? Could it help to come up with a less manual process or procedure to design CoT schemes? How? Currently, the manual decomposition / rewriting of the original problem does all the heavy lifting.
4. The notation in Sec. 3.3 is quite tedious to parse - a figure would really help (probably all of Def. 3.2 could easily be shown graphically).
5. Theorem 3.6 is fine (there always exists a transformer that can solve the CoT problem), but what about learnability (via SGD)? How do we find that transformer in practice? Is that always easy or only sometimes, and if so, when?


**Questions:**

1. What can be said theoretically about the complexity of (n,r) consistent CoT schemes? How does the number of CoT steps scale with problem complexity and problem length (e.g. the CoT scheme for multiplication seems to not scale very well).
2. How would one use the insights from the paper to design reasoning systems that can reason in many settings, or even at LLM / Frontier model scale? Currently the CoT format for each task is very different, and it is unclear that there is any synergy between learning different tasks simultaneously with a single model. The very different CoT schemes may even hurt performance when learning many tasks simultaneously.
3. L 186: Why is it theoretically important that the lengths are the same?
4. How can one in general determine if a problem is (n,r) consistent? Only by finding a valid decomposition (CoT scheme)? How does one find n and r in practice?


**Minor comments:**
1. L 133: “Note that we do not define CoT formally” - this is a weakness for a theory paper and should ideally be fixed.
2. L 136 (Problem statement): The text says “performs well“ which is very informal, whereas the mathematical statement seems to say “performs perfectly in all steps (for any deterministic reasoning problem)”. This discrepancy needs to be fixed - either performing well means always the correct $S^{T+1}$, or there is some other error function that measures how well the model performs. Since this is a theory paper, being precise is important.
3. L 170 says that “with only the continuity bias […] it is almost impossible to predict”. The continuity has never been formulated. I would rather rephrase this to: “without any complexity penalty (such as forms of continuity bias, (n,r) consistency, or forms of Occam’s razor) it is impossible to prefer one continuation over any other, and thus generalizing correctly would rely entirely on chance (with vanishingly small probability as the gap between $N$ and $N’$ grows”.
4. L536-539: I would have liked to see that much earlier in the paper (e.g., around L176).
5. L 251: $s_{j,l}$ and $s_{j,r}$ swapped.
6. L 347: “We use 4 classes” - should be 5.

---

> ### Author Response · Authors · 2024-11-22
> **Response 1 to Reviewer sCy1**
>
> $\textbf{Dear Reviewer sCY1,}$
>
> We would like to express our sincere thanks for your constructive and in-depth review and advice, which we have followed in revising our paper. Specifically, we have reoriented the writing to focus on the empirical contributions. We have strong results that have never been achieved before and we also provide proofs for the solution, which again has not been done before. If you have additional questions or concerns, we will be very happy to address them. Below, we address your comments and answer your questions.
>
> $\textbf{W1}$: The CoT decomposition does most of the heavy lifting ...... relates to general function and complexity classes (e.g., regular languages)
>
> $\textbf{Response}$: We do not have a general and practical way (see our response to question 3) that can be used for all problems. We believe that it is probably unlikely there is a general bias that can be applied to solve all reasoning problems because they are so different from each other, e.g., the calculations involved in addition and multiplication are entirely different. LG is an extrapolation problem and require the learning of underlying rules of calculation of each problem rather than short-cut regularities in the training data, which are only sufficient for interpolation.
>
> For humans, when we teach children to do addition or multiplication, we
> teach them the computation rules rather than letting them guess based
> on the input and output alone. The tags in the CoT scheme of each problem basically inform the learner what elements or positions are involved in the calculation in each step of the problem. This enables the learner to learn the rules of calculation to achieve LG.
>
> Regarding complexity classes, we thought about them but did not have a mature idea. Thus, we wanted this paper to focus on the LG problem alone. Thanks for your advice given in your first point of the ``Improvements'' section below. We followed it and reoriented the writing of the paper to make it focus on the empirical side of the contribution. That is, the paper defines a problem class for which we proposed a condition for its CoT schemes to achieve LG. We believe that our work is still a strong contribution compared to the existing approaches as none of them can solve the multiplication problem perfectly on the tested problem lengths and none have even attempted the division problem. Additionally, none of them has any proof of their solutions.
>
> The changes made to the paper can be found in the abstract and introduction and in lines 137-139 and footnote 3, which are all highlighted in blue. Minor edits are also made in numerous other places to align with the new focus, but not highlighted.
>
> $\textbf{W2}$: Writing throughout most of the manuscript makes claims ...... I will be more specific in the Questions section.
>
> $\textbf{Response}$: There may be some misunderstanding here. We have not claimed that $(n, r)$-consistency is necessary. We only claimed that it is a sufficient condition. As described in the previous response, we have reoriented our paper to focus on the empirical contributions.
>
> $\textbf{W3}$: Learnability and generalization with transformers in practice remain unclear ...... reaching the scale of modern frontier models?
>
> $\textbf{Response}$: As we discussed in footnote 9, this work has not studied learnability. Our theory and proof show the existence of a solution, and our empirical results running on a transformer give perfect accuracy on the tested lengths. Regarding the scale of CoT steps, please refer to our response to your question 1. Regarding learning all tasks simultaneously, please refer to our response to your question 2.
>
> $\textbf{I1}$: A significant overhaul / rewrite to emphasize that the main contribution is not a theory of length generalization and identification of a novel root cause (the root cause is well known in machine learning, mathematics, and philosophy), but a definition of a problem class that has favorable properties w.r.t. length generalization built in, and these properties can be exploited by transformers.
>
> $\textbf{Response}$: Thanks for the valuable advice. We have reoriented the writing to shift the focus of our paper to the empirical contributions. See our response to the first weakness.
>
> Based on the strong empirical contribution, we hope it is sufficient for you to raise your score, as none of the existing approaches in the literature can solve multiplication on the tested problem lengths and none has attempted division. Furthermore, none of the existing papers has any proof of their approaches.
>
> $\textbf{I2}$: If the focus of a major revision lies on the theory ...... what are counterexamples?
>
> $\textbf{Response}$: See our response to the last point above.

---

> > ### Comment · Reviewer_sCy1 · 2024-11-25
> > **Thank you for the detailed response and updated manuscript**
> >
> > I want to thank the authors for their detailed clarifications and improvements to the revised manuscript. Accordingly, I have raised my score and am now weakly in favor of accepting the paper. I still believe some of my criticism remains, and I appreciate that the authors have acknowledged the points raised, though some have not been fully addressed (but as I said I consider them sufficiently addressed; though a stronger version of the work would still be possible).
> >
> > The main improvement is that the writing is now more precise and states the main contribution more clearly (definition of a problem class with a theoretical guarantee), and also emphasises the empirical focus (based on a theoretical understanding, but leaving open some important theoretical questions). Thank you for clarifying that there is no claim that (n,r) consistency is necessary to solve LG. Regarding learnability: I was aware of the footnote in the original manuscript, and I do agree that this is a hard question to address (and the rebuttal phase is not enough for that) - it would make the paper stronger though. Similarly, more results that relate (n,r) consistency to other known complexity classes would make the paper even stronger, as well as a better understanding / more practical method to perform the CoT decomposition. No need to respond to these points again - I am merely explaining why I still do not give the paper a higher score.
> >
> > Finally: "Based on the strong empirical contribution, we hope it is sufficient for you to raise your score, [...]", I certainly appreciate the strong empirical contribution (as I stated in my original review), but I'm afraid that it is mainly due to hand-crafting the CoT schemes, and it is unclear (at least to me) how much the theory and (n,r) consistency have helped here.

---

> > > ### Author Response · Authors · 2024-11-25
> > > **Thank you**
> > >
> > > Thank you very much for raising your score. We greatly appreciate it. We are also grateful for highlighting those theoretical directions to enhance our work further. They provide valuable research challenges for future exploration.
> > >
> > > Thank you once again.
> > >
> > > The Authors

---

> ### Author Response · Authors · 2024-11-22
> **Response 2 to Reviewer sCy1**
>
> $\textbf{I3}$: Strengthen the point of how the current theoretical understanding can help design learnable CoT schemes that generalize. The empirical examples in the paper are good, but how much did the theoretical understanding contribute? Could it help to come up with a less manual process or procedure to design CoT schemes? How? Currently, the manual decomposition / rewriting of the original problem does all the heavy lifting.
>
> $\textbf{Response}$: Yes, we have a method capable of automatically designing CoT schemes to $(n,r)$-consistency. However, its computational complexity is prohibitively high, making it impractical for real-world use. This method is outlined in Appendix B.
>
> $\textbf{I4}$: The notation in Sec. 3.3 is quite tedious to parse - a figure would really help (probably all of Def. 3.2 could easily be shown graphically).
>
> $\textbf{Response}$: Thanks for the suggestion. We thought about it and found it quite hard to draw a figure. But we will keep thinking. If we find a good way, we will add it to the final revised paper.
>
> $\textbf{I5}$: Theorem 3.6 is fine (there always exists a transformer that can solve the CoT problem), but what about learnability (via SGD)? How do we find that transformer in practice? Is that always easy or only sometimes, and if so, when?
>
> $\textbf{Response}$: As we stated in footnote 8 in the revised paper (footnote 4 in the original submission), this paper does not study learnability. Our experiments show that Transformer-based models can learn to achieve LG for challenging mathematical reasoning tasks. We find the Transformer by training with Adam. In experiments, we can always find the Transformer to achieve LG on the tested lengths given the same hyperparameters (e.g. learning rate, train/test length, etc).
>
> $\textbf{Q1}$: What can be said theoretically about the complexity of (n,r) consistent CoT schemes? How does the number of CoT steps scale with problem complexity and problem length (e.g. the CoT scheme for multiplication seems to not scale very well).
>
> $\textbf{Response}$: We still do not have a mature understanding of the complexity of (n, r)-consistent.
>
> Regarding the number of CoT steps, we think it is not a function of $(n, r)$ by the following example. For a problem, let $S_0, S_1, \dots, S_T$ to be the CoT process. We could simply define a new problem by letting $S_0, S_1$ to be the CoT process and $S_1$ to be the final answer. This new problem shares the same (n, r)-consistent property but only has one CoT step for all problem instances.
>
> Regarding the complexity, we conjecture that the complexity is a problem-specific function of the length, which positively correlates to $(n, r)$ and the number of steps, e.g. $C = C(N) \propto n \cdot \log T$, where $C$ is complexity, $N$ is length, $T$ is the number of steps. Further research is still needed.
>
> $\textbf{Q2}$: How would one use the insights from the paper to design reasoning systems that can reason in many settings, or even at LLM / Frontier model scale? Currently the CoT format for each task is very different, and it is unclear that there is any synergy between learning different tasks simultaneously with a single model. The very different CoT schemes may even hurt performance when learning many tasks simultaneously.
>
> $\textbf{Response}$: Yes, for different problems their CoT schemes are different because the calculation operations are quite different for different problems as we explained in the response to the first weakness and in Section 3.5.
>
> We don't expect different tasks to be learned simultaneously. Our vision is that only the basic reasoning functions need to be learned in the proposed way individually. Those more complex reasoning tasks are simply combinations of the basic functions, for which we can make the LLMs/Frontier models to call the basic functions when needed. The current approach of using ever large amount of data to train LLMs may not be the best solution for solving reasoning tasks.
>
> $\textbf{Q3}$: L 186: Why is it theoretically important that the lengths are the same?
>
> $\textbf{Response}$: This is due to the requirement in (iv) of Definition 3.2 for (n, r)-consistency. See lines 259-262.
>
> $\textbf{Q4}$: How can one in general determine if a problem is (n,r) consistent? Only by finding a valid decomposition (CoT scheme)? How does one find n and r in practice?
>
> $\textbf{Response}$: We discussed this issue in Section 3.5. We still do not have a general procedure for these. The general idea is to use tags to let the system know what elements are involved in each reasoning or calculation step such that there is no ambiguity. Also, see our responses to your first point of Weaknesses and the third point of Improvements.

---

> ### Author Response · Authors · 2024-11-22
> **Response 3 to Reviewer sCy1**
>
> $\textbf{M1}$: L 133: “Note that we do not define CoT formally” - this is a weakness for a theory paper and should ideally be fixed.
>
> $\textbf{Response}$: We have defined CoT in lines 122-126.
>
> $\textbf{M2}$: L 136 (Problem statement): The text says “performs well“ which is very informal ...... being precise is important.
>
> $\textbf{Response}$: Fixed. Changed to ``perform perfectly.''
>
> $\textbf{M3}$: L 170 says that “with only the continuity bias […] it is almost impossible to predict” ...... as the gap between $N$ and $N'$  grows”.
>
> $\textbf{Response}$: We rewrite Theorem 3.1.
>
> $\textbf{M4}$: L536-539: I would have liked to see that much earlier in the paper (e.g., around L176).
>
> $\textbf{Response}$: Thanks for the suggestion. It has been done. See footnote 4 in the revised paper.
>
> $\textbf{M5}$: L 251: $s_{j,l}$ and $s_{j,r}$ swapped.
>
> $\textbf{Response}$: We believe that it is correct.
>
> $\textbf{M6}$:  L 347: “We use 4 classes” - should be 5.
>
> $\textbf{Response}$: Fixed.

---

### Official Review · Reviewer_F4Li · 2024-11-06

**Soundness:** 3
**Presentation:** 3
**Contribution:** 2
**Rating:** 5
**Confidence:** 3

**Summary:**

This paper proposes a framework for studying the length generalization ability of scratchpad/chain-of-though (CoT) methods for reasoning problems such as arithmetic. More precisely, a condition, $(n, r)-consistency$, has been proposed such that if CoT satisfies this property, one can design a model capable of length generalization (an approximation/expressivity result). The paper further provides experimental evidence showing that the proposed CoTs are indeed learnable.

**Strengths:**

- The paper tackles the important problem of length generalization on reasoning tasks. It provides a condition for CoTs such that if that condition is satisfied length generalization becomes potentially possible. Theoretical results on the expressively and empirical results on learning are provided in the support of that condition.
- The experimental results seem very interesting and strong (e.g., division is a newly considered task) although there are concerns (see below).
- The multi-line scratchpad/CoT and its implementation are interesting.

**Weaknesses:**

- One of the main weaknesses is the modeling assumption. Generally, the CoT setting works as follows that when we have given $S_0$ as input to the language model (LM), the LM would generate a sequence of thoughts, $S_1, S_2, \ldots, S_T$ that would solve the task (answer is often in $S_T$). However, the modeling in this paper is that we give $S_0$ to the model and get $S_1$ then we give $S_1$ as input to the model to get $S_2$ and so on. So instead of $model(S_0)=S_1, S_2, \ldots, S_T$ we have $model(S_i)=S_{i+1}$. I think this modeling is not a significant issue, it is just an unconventional way (which may work better that the the conventional way) that should be properly acknowledged, explained, and clarified throughout the paper. Currently the paper doesn't acknowledge this difference with normal CoT methods. Further, this problem becomes more significant in two ways:
     - The model doesn't learn the termination condition. In other words, it's not the model that understands $model(S_{T-1})=S_{T}$ is the final step and it shouldn't compute $model(S_T)$, but this is manually controlled by the user.
     - It seems some further processing are done on the inputs and outputs. In particular, it seems input[k+1] is not exactly output[k].

All of these processings and manual handlings seem like engineering hacks which would not generalize to other problems. (Although length generalization on tasks such as addition are important as evaluation metrics, however, we are more generally looking for approaches that would generalize to other problems as well -- in particular to problems that we have less control over the data.)

**Questions:**

- From what I understand from Appendix D, we see that output[k] is not always exactly equal to input[k+1] (e.g., in the addition examples). I think this is some sort of processing that is not necessarily generalizable to other tasks because it uses the knowledge of the task too much (I personally see it an engineering hack). Is there a way to avoid that and be more agnostic towards the task?
- Similar to the concern above, In Appendix D, we see some orange lines that are not predicted by the model and are manually inserted. I think this again reduces the learning part of the paper and is an engineering trick. Can it be avoided?
- I'm not sure if I completely understand how the padding works. When you pad, you still have blank tokens at the start and end of the sequence. However, they are not the middle of any interval. So how are they predicted? Also are the padding tokens provided for all the examples in Appendix D?
- In experiments, the length generalization ability is tried on different lengths and for all of them we see 100% performance. So I wonder what's the point that we see some reduction in the performance? Why haven't you tested the length generalization ability on longer sequences?
- The experiments are done with specialized Transformers. I wonder what would be the length generalization performance if CoT steps were learned by a typical model, e.g., GPT2 or llama?

Minor feedbacks:
- From what I understand, it's not important where the intervals other than the anchor interval are located, is that right? I think it would be  nice if this is further clarified in the paper.
- I think Theorem 3.1 is doing more harm than good. I think the statement that is saying "we have infinite continuations" is more or less intuitive. Further, I think modeling discrete token space with continuous intervals is not justified. Also, why is Lipschitz assumption on these intervals a reasonable property? (and in that case what are the Lipschitz constants?) By default, we expect the distance of the tokens to be more or less the same, however, the Lipschitz assumption goes hand in hand with the Euclidean distance on the intervals which is problematic. For example, if tokens A, B, C correspond to points -1, 0, 1 on the interval. The Lipschitz assumption implies that the semantic distance between (A, B), (B, C) is smaller than (A,C). So I think the Lipschitz property and the use of continuous intervals in this Theorem is unjustified. I would really suggest to use a simpler theorem on discrete token space.
- The abstract and intro imply that authors have found a necessity condition for length generalization and thus I think they should be revised.
- I find the proof sketch part very interesting, and I'd suggest authors to further elaborate on it. (In that case, you may want to move some of the experiment details to the appendix).

---

> ### Author Response · Authors · 2024-11-22
> **Response 1 to Reviewer F4Li**
>
> $\textbf{Dear Reviewer F4Li}$,
>
> Thank you very much for your valuable and constructive comments and questions. Below, we address them point by point. We have updated the paper to reflect the changes.
>
> $\textbf{W1}$: One of the main weaknesses is the modeling assumption ...... Further, this problem becomes more significant in two ways:
>
> $\textbf{Response}$: Thanks for the suggestion. We have acknowledged the difference from the original CoT modeling in footnote 2 in line 160.
>
> $\textbf{W2}$: The model doesn't learn the termination condition ...... but this is manually controlled by the user.
>
> $\textbf{Response}$: Thanks for your suggestion. We have added one special 'EOS' token at the end of the CoT process for each training instance in the dataset. In testing, the output generation process terminates iff 'EOS' is predicted. We ran the experiments again and the results remain the same. We reflected this change in line 451 and line 502 in the revised paper.
>
> $\textbf{W3}$: It seems some further processing are done on the inputs and outputs. In particular, it seems input[k+1] is not exactly output[k].
>
> $\textbf{Response}$: Yes. In each step, the input and output elements need to be aligned as required by $(n,r)$-consistency (see (iv) in Definition 3.2 in line 259). For the arithmetic problem, after a step, empty elements are removed and for addition, an empty element is added for padding (see more details about padding below).
>
> $\textbf{W4}$: All of these processings and manual handlings seem like ...... in particular to problems that we have less control over the data.)
>
> $\textbf{Response}$: We have a different opinion here. We would not regard the proposed method as an engineering hack as we have a theory and proof to support the method. The proposed $(n, r)$-consistency is a general condition. If the CoT scheme of any problem meets the condition, it can achieve LG. $(n, r)$-consistency is not specific to the problems used in the paper because our proofs are not specific to these problems. We believe that existing methods are mainly heuristics as they have no proof of the generality of their methods. Their methods are also much weaker than ours and no paper has attempted the division problem.
>
> $\textbf{Q1}$: From what I understand from Appendix D, we see that output[k] is not always exactly equal to input[k+1] (e.g., in the addition examples). I think this is some sort of processing that is not necessarily generalizable to other tasks because it uses the knowledge of the task too much (I personally see it an engineering hack). Is there a way to avoid that and be more agnostic towards the task?
>
> $\textbf{Response}$: See our response to ''the processing'' above. Regarding the use of the knowledge of the task, we think it is unavoidable because to achieve LG, the learner has to learn the rules of computation for each problem in order to extrapolate rather than short-cut regularities in the training data, which may be enough for interpolation but not enough for extrapolation or LG. That is why we require the CoT scheme of the reasoning problem to satisfy $(n,r)$-consistency. Designing a CoT scheme to meet $(n,r)$-consistency needs the calculation knowledge of the problem. Intuitive, Theorem 3.1 shows that a higher dimensional function is not predictable with only partial observation of the function projected onto a lower dimension. It is thus necessary to have a bias that limits the hypothesis space of the higher dimensional function. We think different problems may need different biases. But it may be possible to have a more general condition than $(n, r)$-consistency.
>
> As humans, when we teach children to do addition or multiplication, we teach them the rules of computation rather than letting them guess based on the input and output alone. The tags in the CoT scheme serve to inform the learner what elements or positions are involved in the computation in each calculating step (also see Section 3.5 in the revised paper.).
>
> That being said, we do have a method capable of automatically designing CoT schemes to satisfy $(n,r)$-consistency. However, its computational complexity is prohibitively high, making it impractical for real-world use. This method is outlined in Appendix B.
>
> $\textbf{Q2}$: Similar to the concern above, In Appendix D, we see some orange lines that are not predicted by the model and are manually inserted. I think this again reduces the learning part of the paper and is an engineering trick. Can it be avoided?
>
> $\textbf{Response}$: These orange lines are only for easy reading or understanding. They are static and pure input and do not change during the CoT process. See examples for multiplication.

---

> ### Author Response · Authors · 2024-11-22
> **Response 2 to Reviewer F4Li**
>
> $\textbf{Q3}$: I'm not sure if I completely understand how the padding works. When you pad, you still have blank tokens at the start and end of the sequence. However, they are not the middle of any interval. So how are they predicted? Also are the padding tokens provided for all the examples in Appendix D?
>
> $\textbf{Response}$: The padding tokens (or elements) at the start and the end are not predicted. They are added to ensure that every predicted token can be the central or middle token of a r-length interval. Since the first token has no token on the left, then it cannot be the central/middle token of an interval. To resolve it, we add about $r/2$ padding tokens before the first token. For the same reason, we add some padding tokens after the end token. They are not provide in examples.
>
> $\textbf{Q4}$: In experiments, the length generalization ability is tried on different lengths and for all of them we see 100\% performance. So I wonder what's the point that we see some reduction in the performance? Why haven't you tested the length generalization ability on longer sequences? The experiments are done with specialized Transformers. I wonder what would be the length generalization performance if CoT steps were learned by a typical model, e.g., GPT2 or llama?
>
> $\textbf{Response}$: We have run experiments on Parity-[2] with length 1000 and still got 100\% accuracy. This is not surprising as Theorem 3.5 shows the existence of a 'perfect' Transformer. We also ran Addition-[2], which gives 100\% accuracy on length 20 (used in the paper), 30, 40, but 73\% on length 50. We think the length amplified the mismatch between the 'learned' Transformer and the 'perfect' Transformer. We will conduct comprehensive experiments on all problems, including multiplication and division, to find the longest lengths that can achieve 100\% accuracy and report in the final revised paper.
>
> Regarding using a typical model, our Transformer adds a relative encoder with mask, which depends on $(n, r)$-consistency property of the reasoning problem (Sec.4.3, 1st paragraph). This mask ensures that every r-length interval is 'clean' i.e. without mixing with noisy tokens beyond the r-length interval. The proof of Theorem 3.5 is a process of matching a sequence of 'clean' r-length intervals. If replacing our Transformer with a typical one, with high probability, the LG would decay a lot as there is no 'clean' r-length interval and the matching process can be chaotic.
>
> $\textbf{M1}$: From what I understand, it's not important where the intervals other than the anchor interval are located, is that right? I think it would be nice if this is further clarified in the paper.
>
> $\textbf{Response}$: Yes, that is right. We clarified this in footnote 6 (line 268).
>
> $\textbf{M2}$: I think Theorem 3.1 is doing more harm than good ...... I would really suggest to use a simpler theorem on discrete token space.
>
> $\textbf{Response}$: Following your suggestion, we have simplified the theorem using a discrete token space. See lines 149-157. The proof is also updated in Appendix C.
>
> $\textbf{M3}$: The abstract and intro imply that authors have found a necessity condition for length generalization and thus I think they should be revised.
>
> $\textbf{Response}$: We have revised it. See lines 18-20 and the first contribution starting from line 63.
>
> $\textbf{M4}$: I find the proof sketch part very interesting, and I'd suggest authors to further elaborate on it. (In that case, you may want to move some of the experiment details to the appendix).
>
> $\textbf{Response}$: We tried but found that there is not much to add without referring to the full proof in the appendix.

---

> > ### Comment · Reviewer_F4Li · 2024-11-27
> >
> > Thank you for the detailed response.
> > To summarize, my concern is that the approach in this paper is too specific to the studied task, in a sense it highly utilizes the knowledge of the task (in scratchpad design, where to put the padding, nonstandard model, etc.) So I have a few questions:
> > - How does this approach generalize and how do you think your approach will be useful in practice in the future?
> > - Can you train a model by just using some sequence of tokens? (without extra interventions in the process such as adding/removing paddings manually at test time)
> >
> > Another concern of mine is that this works completely ignores the positional embeddings for the analysis. In practice, length generalization won't be possible without suitable positional embeddings. However, this work circumvents this need by using a different form of CoT (where each CoT step is fed to the model again to get next step) and by manual paddings during generation. Do you think your circumventions would generalize in practice?
> >
> > Minor typo: there seems to be a typo in the orange line of parity output[3].

---

> ### Author Response · Authors · 2024-11-29
> **Response to Official Comment by Reviewer F4Li**
>
> $\textbf{Dear Reviewer F4Li}$,
>
> Thank you very much for your new comments and questions. We are happy to address them.
>
> $\textbf{Q1}$: How does this approach generalize and how do you think your approach will be useful in practice in the future?
>
> $\textbf{Response}$: Yes, it can achieve generalization, but use a different approach than the familiar methods. We believe that only a finite set of basic reasoning functions needs to be learned individually using the proposed method. More complex reasoning tasks can then be addressed as combinations of these basic functions, with the complex tasks invoking the appropriate basic functions as needed. We suspect this mirrors how humans learn and execute reasoning tasks. Relying on ever-growing amounts of sequential data to train large models in a brute-force manner may not be the most effective solution.
>
> $\textbf{Q2}$: Can you train a model by just using some sequence of tokens? (without extra interventions in the process such as adding/removing paddings manually at test time)
>
> $\textbf{Response}$: We cannot, as certain problems (e.g., arithmetic, addition) cannot guarantee $(n,r)$-consistency without them. It remains unclear whether these problems can achieve $(n,r)$-consistency through some transformations that avoid adding or removing padding. We leave this question for future work.
>
> $\textbf{Q3}$: Another concern of mine is that this works completely ignores the positional embeddings ...... Do you think your circumventions would generalize in practice?
>
> $\textbf{Response}$: Manipulating positional embeddings as input cannot guarantee length generalization (LG), even though existing heuristic methods show some improvement with carefully designed positional embeddings. This limitation arises because such approaches fall within the scope of Theorem 3.1, where the learner does not encounter positional embeddings of longer sequences in training and, therefore, cannot ensure LG in testing. The positional embedding is thus insufficient. In contrast, our $(n, r)$-consistency-based method only requires the learner to see all $n$ $r$-length intervals to guarantee LG, irrespective of the length of the training or testing sequences. We have theoretically proven the existence of a solution that achieves perfect LG under this framework.
>
> $\textbf{Minor typo}$: there seems to be a typo in the orange line of parity output[3].
>
> $\textbf{Response}$: We have fixed it. Thanks.
>
> $\textbf{In summary}$, our work introduces a theoretical framework for understanding reasoning that extends beyond specific problems or architectures. It also proposes a theoretically grounded solution that utilizes the proven effectiveness of Transformers. While we do not claim that our approach is the optimal solution, we hope it serves to inspire further investigation  guided by principled methodologies.
>
> We understand that you may not fully share our perspective. However, we trust you value the importance of pursuing new directions to advance the field. With the proposed theoretical framework and strong empirical results, we hope you agree that our paper represents a significant contribution.
>
> Thank you very much, and we look forward to hearing your feedback.
>
> Authors

---

> ### Author Response · Authors · 2024-12-02
> **A Reminder about Approaching Deadline**
>
> $\textbf{Dear Reviewer F4Li}$,
>
> With the discussion period nearing its end, we would greatly appreciate your feedback on our responses to your recent questions. If you have any additional questions or comments, we would be more than happy to address them.
>
>
> Thank you in advance,
>
> Authors.

---

### Meta-Review · Area_Chair_nZcD · 2024-12-19

**Metareview:**

The paper generated a lot of discussion. The reviewers liked the empirical results, the theory about the consistency property and that the response where the authors acknowledging that the property is a sufficient (but not necessary) condition for number generlizations.

The reviewers were split about the contribution of the work in terms of generlization and felt that the paper is on the borderline.

I recommend accept hoping that the authors will fix the paper to make their claims clearer and position the work in the context correctly by acknowledging the limitations.

**Additional Comments On Reviewer Discussion:**

The paper was discussed quite broadly. AyZa was quite positive about the paper by pointing out to a well rounded paper. While the other reviewer was a bit skeptical of the claims, it was clear that the empirical evaluation made them comfortable with a weak accept.

We request the authors to please take the reviews and discussions into consideration and improve the final version of the paper.

---

### Decision · Program_Chairs · 2025-01-22

Accept (Poster)